# DiSK: Differentially Private Optimizer with Simplified Kalman Filter for Noise Reduction

**Xinwei Zhang**
University of Southern California
xinweiz@usc.edu

**Zhiqi Bu**
Amazon AI
woodyx218@gmail.com

**Borja Balle**
Google DeepMind
bballe@google.com

**Mingyi Hong**
University of Minnesota
mhong@umn.edu

**Meisam Razaviyayn**
University of Southern California
razaviya@usc.edu

**Vahab Mirrokni**
Google Research
mirrokni@google.com

## ABSTRACT

Differential privacy (DP) offers a robust framework for safeguarding individual data privacy. To utilize DP in training modern machine learning models, differentially private optimizers have been widely used in recent years. A popular approach to privatize an optimizer is to clip the individual gradients and add sufficiently large noise to the clipped gradient. This approach led to the development of DP optimizers that have comparable performance with their non-private counterparts in fine-tuning tasks or in tasks with a small number of training parameters. However, a significant performance drop is observed when these optimizers are applied to large-scale training. This degradation stems from the substantial noise injection required to maintain DP, which disrupts the optimizer's dynamics. This paper introduces DiSK, a novel framework designed to significantly enhance the performance of DP optimizers. DiSK employs Kalman filtering, a technique drawn from control and signal processing, to effectively denoise privatized gradients and generate progressively refined gradient estimations. To ensure practicality for large-scale training, we simplify the Kalman filtering process, minimizing its memory and computational demands. We establish theoretical privacy-utility trade-off guarantees for DiSK, and demonstrating provable improvements over standard DP optimizers like DPSGD in terms of iteration complexity upperbound. Extensive experiments across diverse tasks, including vision tasks such as CIFAR-100 and ImageNet-1k and language fine-tuning tasks such as GLUE, E2E, and DART, validate the effectiveness of DiSK. The results showcase its ability to significantly improve the performance of DP optimizers, surpassing state-of-the-art results under the same privacy constraints on several benchmarks.

## 1 INTRODUCTION

Data privacy has become one of the major concerns in modern machine learning systems. Differential Privacy (DP), with its rigorous mathematical foundation, offers a powerful solution. DP provides a framework for developing training algorithms that safeguard the privacy of individuals' data used to train machine learning models. Among various algorithms, Differentially Private Stochastic Gradient Descent (DPSGD) (Abadi et al., 2016) and its variants (Li et al., 2021; Yu et al., 2021; Tang et al., 2024) have emerged as popular choices for training various models, including computer vision (De et al., 2022) and language models (Bu et al., 2023; 2024). These algorithms inject noise into the training process to guarantee privacy. However, this noise injection often comes at a significant cost to model performance, limiting the widespread adoption of DP optimizers (Jayaraman & Evans, 2019). For example, (McMahan et al., 2018; De et al., 2022) observed that DP training led to a $15\%$ drop in model accuracy on the Reddit dataset and a $30\%$ drop on CIFAR-10 compared to non-private training. This challenge has limited the application of DP optimizers primarily to small models or parameter-efficient fine-tuning, as highlighted by (Li et al., 2021).

Many recent works aim to improve differentially private (DP) training performance. These include applying low-pass filters to separate gradients from noise (Zhang et al., 2024a), injecting correlated

noise with algorithms like DP-FTLR (Koloskova et al., 2023; Choquette-Choo et al., 2024), using sharpness-aware minimization for flatter loss landscapes that are less sensitive to DP noise (Park et al., 2023), adaptive clipping (Andrew et al., 2021; Lin et al., 2022; Hu et al., 2021), and model structure optimization (Bu et al., 2024; Papernot et al., 2021; De et al., 2022). However, these methods often require extra memory, lack theoretical guarantees, or have limited applicability. Therefore, there is a strong demand to design a new approach that 1) is memory and computation efficient for implementation, 2) has a theoretical guarantee, and 3) is compatible with a wide range of existing DP optimization algorithms and models. To meet this demand, we leverage the Kalman filter, a tool from control theory, to improve gradient estimates in DP optimization. We further simplify our algorithm for memory and computational efficiency, while maintaining theoretical grounding and broad compatibility with existing DP algorithms and models.

The tools in signal processing and control theory have been leveraged to design novel optimization algorithms. In the context of DP optimization, the error-feedback approach has been adopted to reduce the bias caused by the clipping operation (Zhang et al., 2024b); the low-pass and high-pass filters have been used to separate the gradient from the DP noise (Zhang et al., 2024a; Koloskova et al., 2023; Choquette-Choo et al., 2024); the low-pass spatial filter, including Gaussian and Laplace filters have been used to smooth the privatized model or gradient across dimensions (Wang et al., 2020a; 2021; Liu et al., 2022). These methods effectively improve DP optimizers' performance.

## 1.1 CONTRIBUTIONS

Our approach centers on constructing a dynamic system where the gradient serves as its state. We treat the privatized gradient as a noisy observation of the true gradient within this system. By applying a Kalman filter, we obtain a more accurate estimate of the true gradient by leveraging the gradient dynamics and past estimates, thereby enhancing the performance of DP optimizers. To address the inherent inefficiency of the Kalman filter, we introduce a series of simplifications that reduce memory and computational overhead. Our main contributions can be summarized as follows:

- **Algorithm Design:** We introduce a novel Kalman filter-based approach designed to mitigate DP noise and enhance the performance of various DP optimizers.

- **Algorithm Simplification:** We simplify the Kalman filtering process to significantly reduce memory and computational overhead. This simplified approach, DiSK, requires only one additional forward step and two extra optimizer states.

- **Theoretical Analysis:** We provide theoretical analyses of DiSK, demonstrating that the algorithm is convergent. Moreover, we showed that, compared to DPSGD, DiSK improves the iteration complexity upper-bound by a problem-dependent constant factor.

- **Numerical Results:** Extensive experiments across various models, datasets, and optimizers demonstrate that DiSK significantly boosts DP training performance. Specifically, under the same (or tighter) privacy budgets $\epsilon = 8, \delta = 1/N^{1.1}$, DiSK exhibits substantial improvements in test accuracy for training-from-scratch scenarios: a notable increase from $33.56\%$ to $36.89\%$ on the ImageNet-1k dataset, a considerable rise from $63\%$ to $75\%$ on CIFAR-10, and a remarkable improvement from $21\%$ to $42\%$ on CIFAR-100. Furthermore, in fine-tuning tasks, DiSK demonstrates remarkable improvements: an increase from $85\%$ to $89\%$ on CIFAR-100 and an improvement from $81\%$ to $86\%$ on the GLUE dataset. *These results surpass state-of-the-art DP training performance under the same privacy guarantees.*

## 2 PRELIMINARIES

### 2.1 PROBLEM DEFINITION & NOTATIONS

Typical training procedures require solving the empirical risk minimization (ERM) problem

$$\min_{\mathbf{x} \in \mathbb{R}^d} \left( F(\mathbf{x}) = \frac{1}{N} \sum_{\xi \in \mathcal{D}} f(\mathbf{x}; \xi) \right), \tag{1}$$

where $\mathbf{x} \in \mathbb{R}^d$ is the optimization variable, $\mathcal{D} = \{\xi_1, \ldots, \xi_N\}$ is the training dataset with $|\mathcal{D}| = N$ samples, and $f(\cdot)$ denotes the (possibly non-convex) loss function parameterized by $\mathbf{x}$ and evaluated on sample $\xi$. For solving the above optimization problem, we rely on iterative procedures, such as

SGD, where the parameters are updated over the iterations $t = 1, 2, \ldots$. Throughout the paper, we use $(\cdot)_t$ to denote the variables at iteration $t$, and use $\mathbf{I}_d$ to denote the identity matrix of dimension $d$.

## 2.2 DIFFERENTIALLY PRIVATE OPTIMIZATION

Let us start by recalling the definition of $(\epsilon, \delta)$-Differential Privacy:

**Definition 1 ($(\epsilon, \delta)$-DP (Dwork & Roth, 2014))** *A randomized mechanism $\mathcal{M}$ is said to be $(\epsilon, \delta)$-differentially private if for any two neighboring datasets $\mathcal{D}, \mathcal{D}'$ ($\mathcal{D}, \mathcal{D}'$ differ only by one sample) and for any measurable output set $\mathcal{S}$, it holds that $\Pr[\mathcal{M}(\mathcal{D}) \in \mathcal{S}] \leq e^{\epsilon} \Pr[\mathcal{M}(\mathcal{D}') \in \mathcal{S}] + \delta$.*

A widely used approach to achieving differential privacy (DP) when solving ERM problem (1) is to employ differentially private stochastic gradient descent (Abadi et al., 2016) and its variants, such as DP-Adam and DP-Lora (Yu et al., 2021). To ensure DP, DPSGD leverages the Gaussian mechanism (Dwork & Roth, 2014; Abadi et al., 2016), injecting carefully calibrated Gaussian noise into the gradients at each iteration of the optimization process. This noise injection effectively masks the contribution of individual data points, thereby providing the desired privacy guarantee.

**Definition 2 (Gaussian Mechanism (Dwork & Roth, 2014; Balle & Wang, 2018))** *Suppose an algorithm $\mathcal{A} : \mathcal{D} \rightarrow \mathbb{R}^d$ has $\ell_2$ sensitivity $\Delta_{\mathcal{A}}$, i.e., $\max_{\mathcal{D}, \mathcal{D}'} \|\mathcal{A}(\mathcal{D}) - \mathcal{A}(\mathcal{D}')\| \leq \Delta_{\mathcal{A}}$. Then, for any $\epsilon > 0$ and $\delta \in [0, 1]$, by adding a carefully chosen random Gaussian noise to the output of the algorithm, we can make the algorithm $(\epsilon, \delta)$-DP. More specifically, the mechanism $M(x) = \mathcal{A}(x) + \mathbf{w}$, with $\mathbf{w} \sim \mathcal{N}(0, \sigma_{DP}^2 I_d)$ and $\sigma_{DP}$ satisfies $\Phi\left(\frac{\Delta_{\mathcal{A}}}{2\sigma_{DP}} - \frac{\epsilon \sigma_{DP}}{\Delta_{\mathcal{A}}}\right) - e^{\epsilon} \Phi\left(-\frac{\Delta_{\mathcal{A}}}{2\sigma_{DP}} - \frac{\epsilon \sigma_{DP}}{\Delta_{\mathcal{A}}}\right) \leq \delta$ is $(\epsilon, \delta)$-DP, where $\Phi(t) = \mathbb{P}[\mathcal{N}(0, 1) \leq t]$ is the cumulative density function of normal distribution.*

The DPSGD algorithm, outlined in Algorithm 1, operates by first sampling a mini-batch $\mathcal{B}_t$ of size $B$ and computing the per-sample gradients at each iteration $t$. To guarantee differential privacy, it then applies the Gaussian mechanism, which involves clipping each per-sample gradient to bound its sensitivity to a maximum value $C$ and subsequently injecting Gaussian noise. The clipping operation $\text{clip}(\nabla f, C) =$

---

**Algorithm 1** DPSGD algorithm

**Input:** $\mathbf{x}_0, \mathcal{D}, C, \eta, \sigma_{\text{DP}}$
**for** $t = 0, \ldots, T - 1$ **do**
    Uniformly draw minibatch $\mathcal{B}_t$ from $\mathcal{D}$
    $\mathbf{g}_t = \frac{1}{B} \sum_{\xi \in \mathcal{B}_t} \text{clip}(\nabla f(\mathbf{x}_t; \xi), C) + \mathbf{w}_t$
        where $\mathbf{w}_t \sim \mathcal{N}(0, \sigma_{\text{DP}}^2 \cdot \mathbf{I}_d)$
    $\mathbf{x}_{t+1} = \mathbf{x}_t - \eta_t \mathbf{g}_t,$
**end for**

---

$\min\left\{1, \frac{C}{\|\nabla f\|}\right\} \cdot \nabla f$, often implemented by scaling the gradient when its norm exceeds $C$, limits the influence of any single data point, while the added noise further masks individual contributions, ensuring the desired privacy level (Abadi et al., 2016).

**Theorem 1 (Privacy Guarantee (Abadi et al., 2016))** *Given the number of samples $N$, the batch-size $B$, total number of iterations $T$ and clipping threshold $C$, there exist positive constants $u, v$, such that for any $\epsilon < \frac{uB^2T}{N^2}$ and $0 < \delta$, by choosing $\sigma_{\text{DP}}^2 \geq v \frac{C^2 T \ln(\frac{1}{\delta})}{N^2 \epsilon^2}$, Algorithm 1 is $(\epsilon, \delta)$-DP.*

Theorem 1 implies that the variance of the DP noise injected into the gradients, $\mathbb{E}[\|\mathbf{w}_t\|^2] = d\sigma_{\text{DP}}^2$, scales linearly with both the number of iterations $T$ and the number of parameters $d$. This presents a significant challenge in modern deep learning, where models size $d$ is large (e.g., $22M$-$632M$ for ViT (Dosovitskiy et al., 2020), $137M$-$1.6B$ for GPT-2 (Radford et al., 2019)) and require extensive training (e.g., $300K$ for ViT (Dosovitskiy et al., 2020) and $250K$ for training Llama (Touvron et al., 2023)). Consequently, the magnitude of the injected DP noise can become substantial, leading to a considerable degradation in model performance.

## 2.3 KALMAN FILTER

The Kalman filter is a powerful algorithm that provides estimates of unknown variables by iteratively incorporating a series of measurements over time (Kalman, 1960; Welch et al., 1995). To illustrate its application, let us consider a linear dynamic system characterized by the *System update* and *Observation* equations:

$$
\begin{aligned}
\boldsymbol{\theta}_t &= \mathbf{A}\boldsymbol{\theta}_{t-1} + \mathbf{u}_t + \mathbf{v}_t, & \textit{(System update)} \\
\boldsymbol{\psi}_t &= \mathbf{C}\boldsymbol{\theta}_t + \mathbf{w}_t, & \textit{(Observation)}
\end{aligned}
\tag{2}
$$

where $\mathbf{A} \in \mathbb{R}^{d_\theta \times d_\theta}$, $\mathbf{C} \in \mathbb{R}^{d_\psi \times d_\theta}$ are the transition and observation matrices; $\boldsymbol{\theta}_t \in \mathbb{R}^{d_\theta}$ is the unknown variable to be tracked/estimated; $\boldsymbol{\psi}_t \in \mathbb{R}^{d_\psi}$ denotes the observation of the system; $\mathbf{u}_t \in \mathbb{R}^{d_\theta}$ denotes the known input; and $\mathbf{v}_t \in \mathbb{R}^{d_\theta}$, $\mathbf{w}_t \in \mathbb{R}^{d_\psi}$ are the process and observation noises that follow Gaussian distribution $\mathcal{N}(0, \Sigma_{\mathbf{v}}), \mathcal{N}(0, \Sigma_{\mathbf{w}})$, respectively. Then, the Kalman filter uses the following updates to track $\{\boldsymbol{\theta}_t\}$ (Kalman, 1960; Welch et al., 1995):

$$\tilde{\boldsymbol{\theta}}_{t|t-1} = \mathbf{A}\tilde{\boldsymbol{\theta}}_{t-1} + \mathbf{u}_t \qquad \textit{(Prediction)}$$
$$\mathbf{P}_{t|t-1} = \mathbf{A}\mathbf{P}_{t-1}\mathbf{A}^\top + \Sigma_{\mathbf{v}}$$
$$\mathbf{K}_t = \mathbf{P}_{t|t-1}\mathbf{C}^\top(\mathbf{C}\mathbf{P}_{t|t-1}\mathbf{C}^\top + \Sigma_{\mathbf{w}})^{-1} \qquad (3)$$
$$\tilde{\boldsymbol{\theta}}_t = (\mathbf{I}_{d_\theta} - \mathbf{K}_t\mathbf{C})\tilde{\boldsymbol{\theta}}_{t|t-1} + \mathbf{K}_t\boldsymbol{\psi}_t \qquad \textit{(Correction)}$$
$$\mathbf{P}_t = (\mathbf{I}_{d_\theta} - \mathbf{K}_t\mathbf{C})\mathbf{P}_{t|t-1}.$$

The filter first *predicts* the state $\tilde{\boldsymbol{\theta}}_{t|t-1}$ by system dynamics, and compute the *filter gain* $\mathbf{K}_t \in \mathbb{R}^{d_\theta \times d_\psi}$ based on the covariance matrix $\mathbf{P}_t \in \mathbb{R}^{d_\theta \times d_\theta}$, and *corrects* the prediction with system observation $\boldsymbol{\psi}_t$ to obtain the estimate $\tilde{\boldsymbol{\theta}}_t$ at time $t$. The Kalman filter makes use of both the noisy observation and the prior knowledge of the system dynamics to obtain an accurate estimation of the state $\boldsymbol{\theta}_t$.

A key innovation of this work is to treat the privatized gradients in the DP optimization as noisy observations of the true underlying gradients. By constructing gradient dynamics using Taylor expansion, we establish a framework for applying the Kalman filter to refine these noisy observations and obtain more accurate gradient estimates. To ensure practical applicability for large-scale models, we simplify the Kalman filter, significantly reducing its memory and computational footprint. These improved gradient estimates ultimately lead to enhanced performance in differentially private optimizers.

## 2.4 RELATED WORKS

**Optimization with filters and controllers:** The use of filters and controllers in designing and analyzing optimization algorithms has a rich history. Researchers have leveraged high-pass and low-pass filters to enhance gradient estimation in zeroth-order optimization (Chen et al., 2022), employed PID controllers for both centralized and distributed optimization (Wang et al., 2020b), and analyzed optimizers through the lens of control theory, treating them as dynamic systems (Lessard et al., 2016; Hu & Lessard, 2017; Muehlebach & Jordan, 2019; Mohammadi et al., 2024; Badithela & Seiler, 2019; Cyrus et al., 2018; Scherer et al., 2023; Zhang et al., 2023).

**Kalman filter for optimization:** The Kalman filter has been utilized in convex optimization for reducing stochastic gradient noise (Bittner & Pronzato, 2004; Vuckovic, 2018). Vuckovic (2018) uses the dynamics of the optimization variable and the gradient to construct the Kalman filter to analyze and improve the performance of momentum methods. Bittner & Pronzato (2004) uses gradient and Hessian as its states to construct the dynamic system for SGD to construct a stopping rule. However, these approaches, with their direct application of the Kalman filter, incur prohibitively high computational and memory costs, ranging from $\mathcal{O}(d^3)$ to $\mathcal{O}(d^6)$, rendering them impractical for training large-scale machine learning models.

**Improving DP optimization:** Numerous techniques have been proposed to enhance DP optimization by mitigating the impact of DP noise. These include adaptive gradient clipping methods that dynamically adjust clipping thresholds (Andrew et al., 2021; Bu et al., 2024), parameter-efficient training strategies employing adapters, low-rank weights, or quantization (Yu et al., 2021; Luo et al., 2021; Yu et al., 2021), and the design of specialized model architectures less susceptible to noise perturbations (De et al., 2022; Papernot et al., 2021; Wang et al., 2020a). Furthermore, drawing inspiration from signal processing, researchers have explored the use of colored high-frequency DP noise to separate it from the gradient (Koloskova et al., 2023) and the application of low-pass filters to extract the gradient signal from noisy observations (Zhang et al., 2024a).

## 3 ALGORITHM DESIGN

This section introduces the general Noise Reduction for **Di**fferentially Private Optimizers with **S**implified **K**alman Filter (DiSK) framework. This approach leverages the inherent dynamics of the gradient and employs Kalman filtering to obtain denoised estimates of the true gradients from

their noisy, privatized counterparts. To enhance its practicality for modern deep learning, Section 3.2 details a simplified version of the Kalman filter updates, designed for memory and computational efficiency.

### 3.1 GRADIENT DYNAMIC AND KALMAN FILTER

To explain our idea of using the Kalman filter for denoising the gradient, let us start by first establishing a dynamic system for the gradients, comprising a "system update" equation and an "observation" equation: The **system update** of the gradient dynamics can be derived by the Taylor expansion and quantifying the change of the gradient at iteration $t$:

$$\nabla F(\mathbf{x}_t) = \nabla F(\mathbf{x}_{t-1}) + \mathbf{H}_t \cdot (\mathbf{x}_t - \mathbf{x}_{t-1}) + \mathbf{v}_t, \tag{4}$$

where $\mathbf{H}_t := \nabla^2 F(\mathbf{x}_{t-1}) \in \mathbb{R}^{d \times d}$ and $\mathbf{v}_t = \frac{1}{2} \int_0^1 \nabla^3 F(\mathbf{x}_{t-1})(z\mathbf{x}_t + (1-z)\mathbf{x}_{t-1})^{\otimes 2} \mathrm{d}z$ is the remainder and $(\cdot)^\otimes$ denotes the tensor vector product. The **observation** of the system is defined as the privatized gradient $\mathbf{g}_t$, which is a stochastic mapping of the true gradient:

$$\mathbf{g}_t = \frac{1}{B} \sum_{\xi \in \mathcal{B}_t} \mathrm{clip}\left(\nabla f(\mathbf{x}_t, \xi), C\right) + \mathbf{w}_t = \mathbf{C}_t \nabla F(\mathbf{x}_t) + \mathbf{w}'_t, \tag{5}$$

where $\mathbf{w}'_t$ is the observation noise containing the DP noise and sub-sampling noise; $\mathbf{C}_t$ is the observation matrix. If the clipping operation is inactive and $\mathcal{B} = \mathcal{D}$, i.e., the full batch gradient is used, then $\mathbf{C}_t = \mathbf{I}_d$. Otherwise, $\mathbf{C}_t$ depends on the clipping factor and the mini-batch $\mathcal{B}$. Combining the system update (4) and the observation (5), we can model the system dynamic of the gradient as:

$$\begin{aligned} \nabla F(\mathbf{x}_t) &= \nabla F(\mathbf{x}_{t-1}) + \mathbf{H}_t(\mathbf{x}_t - \mathbf{x}_{t-1}) + \mathbf{v}_t, & \textit{(System update)} \\ \mathbf{g}_t &= \mathbf{C}_t \nabla F(\mathbf{x}_t) + \mathbf{w}'_t. & \textit{(Observation)} \end{aligned} \tag{6}$$

Let us compare our dynamic system with the general one in (2): In our dynamic system, the gradient $\nabla F(\mathbf{x}_t)$ plays the role of the state $\boldsymbol{\theta}_t$. Other parameters in (2) have the following correspondence to gradient dynamics: $\mathbf{A} = \mathbf{I}_d, \mathbf{C} = \mathbf{C}_t$, the input $\mathbf{u}_t = \mathbf{H}_t(\mathbf{x}_t - \mathbf{x}_{t-1})$, and the observation $\boldsymbol{\psi}_t = \mathbf{g}_t$. With the above mapping, we can apply the Kalman filter that combines the **system update** and the **observation** of the gradient to improve the overall estimation quality of the actual gradient beyond only using the observation $\mathbf{g}_t$. However, there are two key challenges when applying Kalman filter to (6): 1) the input $\mathbf{H}_t(\mathbf{x}_t - \mathbf{x}_{t-1})$ is hard to obtain as computing the Hessian matrix $\mathbf{H}_t$ is challenging for large models, and we can only approximate $\mathbf{H}_t(\mathbf{x}_t - \mathbf{x}_{t-1})$, resulting in a noisy input; 2) the observation matrix $\mathbf{C}_t$ is an unknown time-varying random matrix, resulting a multiplicative observation noise. Due to these differences, the traditional Kalman filter (Kalman, 1960) cannot be directly applied. Instead, we apply the Kalman filter with noisy input and multiplicative observation noise proposed in Wu et al. (2016) to our system (6), leading to the update rules:

$$\begin{aligned} \tilde{\mathbf{g}}_{t|t-1} &= \tilde{\mathbf{g}}_{t-1} + \tilde{\mathbf{H}}_t(\mathbf{x}_t - \mathbf{x}_{t-1}) & \textit{(Prediction)} \\ \mathbf{P}_{t|t-1} &= \mathbf{P}_{t-1} + \Sigma_{\mathbf{H}} + \Sigma_{\mathbf{v}} \\[4pt] \mathbf{K}_t &= \mathbf{P}_{t|t-1}\mathbb{E}[\mathbf{C}_t]^\top \left(\Sigma_{\mathbf{w}} + \mathbb{E}[\mathbf{C}_t]\left(\Sigma_{\mathbf{C}}\mathbf{S}_t + \mathbf{P}_{t|t-1}\right)\mathbb{E}[\mathbf{C}_t]^\top - \Sigma_{\mathbf{H}}\right)^{-1} \\ \tilde{\mathbf{g}}_t &= (\mathbf{I} - \mathbf{K}_t\mathbb{E}[\mathbf{C}_t])\tilde{\mathbf{g}}_{t|t-1} + \mathbf{K}_t\mathbf{g}_t & \textit{(Correction)} \\ \mathbf{P}_t &= (\mathbf{I} - \mathbf{K}_t\mathbb{E}[\mathbf{C}_t])\mathbf{P}_{t|t-1} \\ \mathbf{S}_t &= \mathbb{E}[\tilde{\mathbf{g}}_t\tilde{\mathbf{g}}_t^\top], \end{aligned}$$

where $\mathbf{P}_t$ denotes the covariance matrix of $\tilde{\mathbf{g}}_t$, $\Sigma_{\mathbf{H}}, \Sigma_{\mathbf{w}}, \Sigma_{\mathbf{C}}, \Sigma_{\mathbf{v}}$ denote the covariance matrices of the random variables $\mathbf{H}_t, \mathbf{w}_t, \mathbf{C}_t, \mathbf{v}_t$, respectively, and $\tilde{\mathbf{H}}_t$ is an instantiation/observation of the unknown Hessian matrix $\mathbf{H}_t$. The difference between the above Kalman filter and the original Kalman filter (3) is highlighted in magenta color. The variance of the Hessian $\tilde{\mathbf{H}}_t$ plays a role in updating $\mathbf{P}_{t|t-1}, \mathbf{K}_t$, and the expectation $\mathbb{E}[\mathbf{C}_t]$ of the random observation matrix $\mathbf{C}_t$ is used for the updates and its variance $\Sigma_{\mathbf{C}}$ also appears in the update of $\mathbf{K}_t$. Compared to the noisy gradient $\mathbf{g}_t$, the output of the Kalman filter $\tilde{\mathbf{g}}_t$ has a smaller variance, resulting in improved performance when used in DP optimizers.

The resulting optimizer with the Kalman filter is given in Algorithm 2. The hyper-parameters of the Kalman filter are the expected observation matrix $\mathbb{E}[\mathbf{C}]$ and the variances of the noises in the system, i.e., $\sigma_{\mathbf{w}}^2, \Sigma_{\mathbf{C}}, \Sigma_{\mathbf{H}}$. Here, we dropped subscript $t$ for $\mathbf{C}_t$ for simplicity in our modeling.

---

**Algorithm 2** Optimizer with Kalman Filter

---

1: **Input:** $\mathbf{x}_0, \mathcal{D}, \eta, \mathbb{E}[\mathbf{C}], \sigma_{\mathbf{w}}^2, \Sigma_{\mathbf{C}}, \Sigma_{\mathbf{H}}$
2: **Initialize:** $\tilde{\mathbf{g}}_{-1} = 0, \mathbf{d}_{-1} = 0, \mathbf{P}_{-1} = \sigma_{\mathbf{w}}^2 \mathbf{I}_d$
3: **for** $t = 0, \ldots, T - 1$ **do**
4:      Randomly draw minibatch $\mathcal{B}_t$ from $\mathcal{D}$
5:      Compute privatized gradient $\mathbf{g}_t = \frac{1}{B} \sum_{\xi \in \mathcal{B}_t} \text{clip}\left(\nabla f(\mathbf{x}_t; \xi), C\right) + \mathbf{w}_t$        *# Gradient observation*
6:      $\tilde{\mathbf{g}}_{t|t-1} = \tilde{\mathbf{g}}_{t-1} + \mathbf{H}_t \mathbf{d}_{t-1}$        *# Prediction*
7:      $\mathbf{P}_{t|t-1} = \mathbf{P}_{t-1} + \Sigma_{\mathbf{H}}$
8:      $\mathbf{K}_t = \mathbf{P}_{t|t-1} \mathbb{E}[\mathbf{C}]^\top \left(\mathbb{E}[\mathbf{C}](\mathbf{P}_{t|t-1} + \Sigma_{\mathbf{C}} \tilde{\mathbf{g}}_t \tilde{\mathbf{g}}_t^\top) \mathbb{E}[\mathbf{C}]^\top + \sigma_{\mathbf{w}}^2 \mathbf{I}_d - \Sigma_{\mathbf{H}}\right)^{-1}$   *# Compute gain*
9:      $\tilde{\mathbf{g}}_t = \tilde{\mathbf{g}}_{t|t-1} + \mathbf{K}_t (\mathbf{g}_t - \mathbb{E}[\mathbf{C}]\tilde{\mathbf{g}}_{t|t-1})$        *# Compute denoised private gradient*
10:     $\mathbf{x}_{t+1} = \text{OptimizerUpdate}(\mathbf{x}_t, \eta, \tilde{\mathbf{g}}_t)$        *# Parameter update*
11:     $\mathbf{d}_t = \mathbf{x}_{t+1} - \mathbf{x}_t$        *# Record update direction*
12:     $\mathbf{P}_t = (\mathbf{I} - \mathbf{K}_t \mathbb{E}[\mathbf{C}])\mathbf{P}_{t|t-1}$        *# Update covariance matrix*
13: **end for**

---

## 3.2 Algorithm simplification

While Algorithm 2 provides a general framework for applying Kalman filtering to various optimizers, it faces significant challenges in terms of computational and memory demands. Specifically, accurately computing the Hessian matrix (**H**) for large models under DP constraints is infeasible, the matrix inversion step introduces a cubic computational complexity ($\mathcal{O}(d^3)$), and storing the covariance matrix ($\mathbf{P}_t$) at each iteration requires quadratic memory ($\mathcal{O}(d^2)$). To address these limitations and arrive at a practical and memory-efficient algorithm, we propose the following simplifications:

**Constant $\mathbf{C}_t$:** We simplify the random matrix $\mathbf{C}_t$ to an identity matrix $\mathbf{I}_d$, and simply model the randomness in the observation as additive noise only. This simplification is achieved in two steps corresponding to the two sources of randomness in $\mathbf{C}_t$. First, we remove the impact of clipping in $\mathbf{C}_t$. This is justified when 1) the clipping threshold $C$ is large enough so that clipping is inactive; or when 2) the clipped gradient $\nabla F_C(\mathbf{x})$ has zero curl, so that our method optimizes $F_C(\mathbf{x})$, where

$$F_C(\mathbf{x}) = \int_0^1 \nabla F_C(z\mathbf{x})^\top \mathbf{x} \mathrm{d}z, \ \nabla F_C(\mathbf{x}) = \frac{1}{N} \sum_{\xi \in \mathcal{D}} \text{clip}\left(\nabla f(\mathbf{x}; \xi), C\right). \tag{7}$$

Under this assumption, the minibatch clipped gradient $\frac{1}{B} \sum_{\xi \in \mathcal{B}} \text{clip}\left(\nabla f(\mathbf{x}, \xi), C\right)$ is an unbiased estimation of $\nabla F_C(\mathbf{x})$. Thus, $\mathbb{E}[\mathbf{C}_t] = \mathbf{I}_d$. Second, the randomness of sub-sampling in $\mathbf{C}_t$ is removed by assuming that the sub-sampled mini-batch gradient only causes additive noise, i.e., $\frac{1}{B} \sum_{\xi \in \mathcal{B}} \text{clip}\left(\nabla f(\mathbf{x}, \xi), C\right) = \nabla F_C(\mathbf{x}) + \mathbf{w}_{\text{SGD}}$, so that $\mathbf{C} = \mathbf{I}_d$ and $\Sigma_{\mathbf{C}} = 0$.

**Hessian estimation:** We apply the following "trick" to bypass the explicit computation of the Hessian matrix, $\mathbf{H}_t$: we approximate the Hessian-vector product, $\mathbf{H}_t(\mathbf{d}_{t-1})$, using a finite difference method (Pearlmutter, 1994):

$$\mathbf{H}_t \mathbf{d}_{t-1} = \frac{\nabla F(\mathbf{x}_t + \gamma \mathbf{d}_{t-1}) - \nabla F(\mathbf{x}_t)}{\gamma} + \mathcal{O}(\gamma) \approx \frac{1}{B} \sum_{\xi \in \mathcal{B}} \frac{\nabla f(\mathbf{x}_t + \gamma \mathbf{d}_{t-1}; \xi) - \nabla f(\mathbf{x}_t; \xi)}{\gamma}.$$

This estimation eliminates the need for expensive Hessian computations and significantly reduces both memory and computational complexity.

**Simplification of $\Sigma_H$:** A few matrix computation steps in Algorithm 2 are extremely time-consuming and memory inefficient. Specifically, the matrix inversion in Line 8 of Algorithm 2 incurs a cubic computational cost ($\mathcal{O}(d^3)$), which becomes prohibitive for large models with billions of parameters. Therefore, we simplify the algorithm by assuming that the covariance matrix in the Kalman filter is a time-invariant identity matrix scaled with a constant, i.e., $\Sigma_H = \sigma_H^2 \mathbf{I}_d$. Under this assumption, the matrices $\mathbf{P}_t$ and $\mathbf{K}_t$ become $p_t \mathbf{I}_d, k_t \mathbf{I}_d$, for some scalars $p_t, k_t$. Thus, we reduce all matrix computations to efficient scalar-vector multiplications and the memory complexity storing these matrices to $\mathcal{O}(1)$, making our algorithm a viable option for DP training of large-scale models.

**Filter gain simplification:** With the above simplification, the updates of $p_t, k_t$ simplify to $p_t = \frac{(\sigma_{\mathbf{w}}^2 - \sigma_H^2)(p_{t-1} + \sigma_{\mathbf{v}}^2 + \sigma_H^2)}{p_{t-1} + \sigma_{\mathbf{v}}^2 + \sigma_{\mathbf{w}}^2}, k_t = \frac{p_{t-1} + \sigma_{\mathbf{v}}^2 + \sigma_H^2}{p_{t-1} + \sigma_{\mathbf{v}}^2 + \sigma_{\mathbf{w}}^2}$. Therefore, $k_t$ converges to its stable value with a linear rate, i.e., $\|k_t - k_\infty\| = \mathcal{O}(c_k^t)$, where $c_k = \frac{2\sigma_{\mathbf{w}}^2 + 3\sigma_H^2 + \sigma_{\mathbf{v}}^2 - \sqrt{(\sigma_{\mathbf{v}}^2 + \sigma_H^2)(4\sigma_{\mathbf{w}}^2 + \sigma_{\mathbf{v}}^2 - 3\sigma_H^2)}}{2\sigma_{\mathbf{w}}^2 + 3\sigma_H^2 + \sigma_{\mathbf{v}}^2 + \sqrt{(\sigma_{\mathbf{v}}^2 + \sigma_H^2)(4\sigma_{\mathbf{w}}^2 + \sigma_{\mathbf{v}}^2 - 3\sigma_H^2)}} \in (0, 1)$ (see

---

**Algorithm 3** DiSK: **Di**fferentially private optimizer with **S**implified **K**alman filter

---
1: **Input:** $\mathbf{x}_0, \mathcal{D}, \eta, \gamma, \kappa, C, \sigma_{\mathrm{DP}}$
2: **Initialize:** $\tilde{\mathbf{g}}_{-1} = \mathbf{g}_0, \mathbf{d}_{-1} = 0$
3: **for** $t = 0, \ldots, T-1$ **do**
4:     Randomly draw minibatch $\mathcal{B}_t$ from $\mathcal{D}$
5:     $\mathbf{g}_t = \frac{1}{B} \sum_{\xi \in \mathcal{B}_t} \mathrm{clip} \left( \frac{1-\kappa}{\kappa\gamma} \nabla f(\mathbf{x}_t + \gamma \mathbf{d}_{t-1}; \xi) + (1 - \frac{1-\kappa}{\kappa\gamma}) \nabla f(\mathbf{x}_t; \xi), C \right) + \mathbf{w}_t$
        where $\mathbf{w}_t \sim \mathcal{N}(0, \sigma_{\mathrm{DP}}^2 \cdot \mathbf{I}_d)$
6:     $\tilde{\mathbf{g}}_t = (1-\kappa)\tilde{\mathbf{g}}_{t-1} + \kappa \mathbf{g}_t$                    *# Apply Kalman filter*
7:     $\mathbf{x}_{t+1} = \mathrm{OptimizerUpdate}(\mathbf{x}_t, \eta, \tilde{\mathbf{g}}_t)$             *# Update model*
8:     $\mathbf{d}_t = \mathbf{x}_{t+1} - \mathbf{x}_t$                         *# Record update direction*
9: **end for**

---

derivations in Appendix A.3). So we can use constant $k_t = \kappa, \forall\, t$ to further simplify the algorithm and avoid iteratively updating $p_t$ and recomputing $k_t$ for each step.

With the above simplifications, the complex Algorithm 2 simplifies to DiSK in Algorithm 3. For a detailed walkthrough of this simplification process, please refer to Appendix A.3. This simplified algorithm computes and privatizes the linear combination of the gradient evaluated at two points, $\mathbf{x}_t, \mathbf{x}_t + \gamma \mathbf{d}_t$. Then, this privatized combination undergoes exponential weighted averaging to obtain $\tilde{\mathbf{g}}_t$, which serves as the input to the base optimizer (Line 7 of Algorithm 3).

### 3.3 ADDITIONAL DISCUSSION

**Memory and computation cost:** Compared with the base DP optimizer, Algorithm 3 requires one additional forward step to compute $f(\mathbf{x}_t + \gamma \mathbf{d}_{t-1}; \xi)$. This means that the algorithm has at most twice the computational cost of the baseline DP-SGD algorithm. Moreover, Algorithm 3 only requires two additional states to store: $\tilde{\mathbf{g}}_t$ and $\mathbf{d}_t$. Compared to DPSGD, which requires storing the model and the gradient, DiSK has at most twice the memory cost; and compared to DPAdam, which requires storing the first- and second-order moments, the algorithm has at most $1.5\times$ the memory cost.

**Connection to NAG and STORM:** Remarkably, Algorithm 3 has an implicit connection to the (unified) Nesterov accelerated gradient (NAG) method (Shen et al., 2023; Sutskever et al., 2013) and the Stochastic Recursive Momentum (STORM) algorithm (Cutkosky & Orabona, 2019). Specifically, by letting the OptimizerUpdate be SGD, assuming clipping is inactive, and setting $\mathbf{w}_t = 0$ (i.e., without privatizing the gradient) in Algorithm 3, we can make a clear connection: On one hand, DiSK reduces to NAG by choosing $\gamma = \frac{1-\kappa}{\kappa}, \eta = \kappa, 1 - \kappa = \mu$, and $B = N$. On the other hand, the update of DiSK matches STORM by choosing $\gamma = -1, \kappa = \alpha$, and $B = 1$. A detailed derivation and discussion is provided in Appendix A.4. This observation reveals an intriguing connection between NAG (designed for acceleration) and STORM (focused on variance reduction), unifying them within the framework of DiSK, a Kalman filtering-based algorithm.

**Connection to DOPPLER:** DOPPLER (Zhang et al., 2024a) and DiSK both use a filter to separate the gradient signal from the DP noise. If $\mathbf{g}_t$ only evaluates the gradient at $\mathbf{x}_t$ instead of using the linear combination of gradients at two points, $\mathbf{x}_t, \mathbf{x}_t + \gamma \mathbf{d}_{t-1}$ in Algorithm 3, Line 5, DiSK becomes DOPPLER with a first-order filter. The key difference is that DOPPLER assumes an underlying low-frequency dynamic of the gradient and applies a time-invariant low-pass filter. Designing the optimal low-pass filter relies on the prior knowledge of the gradient frequency spectrum, which is hard to obtain in practice, and implementing such a high-order low-pass filter results in a large memory overhead. While in DiSK, we do not assume the frequency property of the gradient signal. Instead, we incorporate the gradient dynamics into the filtering procedure and use the Kalman filter, a predictive filtering approach, to reduce the impact of DP noise.

## 4 THEORETICAL ANALYSIS

This section includes theoretical analyses of Algorithm 3. Our study establishes the convergence, provable noise reduction, and the privacy-utility trade-off of the algorithm. To facilitate our analysis, we make the following assumptions:

**A 1 (Smoothness)** $f(\cdot, \xi)$ *is* $L$*-smooth for any* $\xi$*, i.e.,* $\|\nabla f(\mathbf{x}; \xi) - \nabla f(\mathbf{y}; \xi)\| \leq L \|\mathbf{x} - \mathbf{y}\|$, $\forall \xi \in \mathcal{D}$, $\forall \mathbf{x}, \mathbf{y} \in \mathbb{R}^d$.

**A 2 (Bounded Variance)** *The per-sample gradient has bounded variance with* $\mathbb{E}_{\xi \in \mathcal{D}} \|\nabla f(\mathbf{x}; \xi) - \nabla F(\mathbf{x})\|^2 \leq \sigma_{SGD}^2$, $\forall \mathbf{x} \in \mathbb{R}^d$, *where* $\mathbb{E}_{\xi \in \mathcal{D}}[\cdot]$ *denotes the expectation taken on the randomness over $\xi$ that is uniformly sampled from dataset $\mathcal{D}$.*

**A 3 (Bounded Gradient)** *Each per-sample gradient has a bounded norm, i.e., $\|\nabla f(\mathbf{x}; \xi)\| \leq G$, $\forall \mathbf{x} \in \mathbb{R}^d, \forall \xi \in \mathcal{D}$.*

Let us briefly comment on these assumptions: A1 and A2 are standard in non-convex optimization (Allen-Zhu & Hazan, 2016; Zaheer et al., 2018; Abadi et al., 2016); and A3 is commonly used in analyzing the convergence of DP algorithms (Abadi et al., 2016; Wang et al., 2020a; Andrew et al., 2021) to avoid introducing the clipping bias. Since the impact of clipping is not the major focus of this paper, we follow this tradition and use A3 to simplify our theoretical analysis.

### 4.1 CONVERGENCE ANALYSIS

We provide the following convergence results for Algorithm 3, assuming $\sigma_{\mathrm{DP}}$ being a constant.

**Theorem 2** *Assume A1-A3 hold. Fix $\sigma_{\mathrm{DP}}^2$ and choose $C \geq (1 + \frac{2(1-\kappa)}{\kappa})G$, $\kappa, \eta$ satisfy*
$$\eta < \frac{1+\kappa}{2L(1 + 2(1-\kappa)^2\beta L(2 + |1+\gamma| M_\gamma))}, \quad \kappa > 1 - \frac{1}{\sqrt{1 + 4\eta^2 L^2 + |1+\gamma|(\kappa + 2\eta^2 L^2 M_\gamma)}},$$
*and run Algorithm 3 for $T$ iterations. Then,*
$$\frac{1}{T}\sum_{t=0}^{T}\mathbb{E}\|\nabla F(\mathbf{x}_t)\|^2 \leq \frac{2(F(\mathbf{x}_0) + \beta\|\nabla F(\mathbf{x}_0)\|^2 - F^\star)}{M_1 \eta T}$$
$$+ \frac{2(\beta + \eta^2 L)\kappa^2}{M_1 \eta}\left(\frac{(2 + |1+\gamma|)\sigma_{\mathrm{SGD}}^2}{B} + d\sigma_{\mathrm{DP}}^2\right), \quad (8)$$
*where $M_\gamma = 1 + \frac{4(2 + 1/\kappa + |1+\gamma|)}{\gamma^2}$, $M_1 = (1 + \kappa - 2\eta L) - 4(\beta + \eta^2 L)(1-\kappa)^2 L^2 \eta(2 + |1+\gamma| M_\gamma) > 0$, and $\beta \geq \frac{\eta(1-\kappa)/2 + \eta^2 L(1-\kappa)^2(1 + 4\eta^2 L^2 + |1+\gamma|(\kappa + 2\eta^2 L^2 M_\gamma))}{1 - (1-\kappa)^2(1 + 4\eta^2 L^2 + |1+\gamma|(\kappa + 2\eta^2 L^2 M_\gamma))} \geq 0$ are some non-negative constants.*

The proof of Theorem 2 is relegated to Appendix B. Notice that when $\kappa = 1$, we have $\beta = 0$, $M_1 = 2(1 - \eta L)$ and the convergence result recovers that of DPSGD (Ghadimi & Lan, 2013; Zhang et al., 2017). With this theorem, we can choose specific parameters and obtain the following corollary:

**Corollary 1** *Under the conditions of Theorem 2, choose $\gamma = -1$,*
$$\eta = \min\left\{\frac{1}{L(2 + 4/M_\kappa - M_\kappa)}, \frac{1}{M_\kappa L}\sqrt{\frac{2M_\kappa L(F(\mathbf{x}_0) - F^\star) + \|\nabla F(\mathbf{x}_0)\|^2}{2Td\sigma_{\mathrm{DP}}^2}}\right\},$$
$$\beta = \frac{\eta(1-\kappa)/2 + \eta^2 L(1-\kappa)^2(1 + 4\eta^2 L^2)}{1 - (1-\kappa)^2(1 + 4\eta^2 L^2)} \leq \frac{1}{2M_\kappa L},$$
$$\kappa = M_\kappa L\eta \leq 1, M_\kappa = \frac{\|\nabla F(\mathbf{x}_0)\|^2}{2L(F(\mathbf{x}_0) - F^\star)} \leq 1, \quad and \quad B \geq \max\left\{1, \frac{2\sigma_{\mathrm{SGD}}^2}{d\sigma_{\mathrm{DP}}^2}\right\},$$
*and sufficiently large $T \geq \frac{2L(F(\mathbf{x}_0) - F^\star)(\frac{16}{M_\kappa^3} + \frac{16}{M_\kappa^2} - \frac{4}{M_\kappa} - 4) + \|\nabla F(\mathbf{x}_0)\|^2}{d\sigma_{\mathrm{DP}}^2}$, then Algorithm 3 satisfies:*
$$\frac{1}{T}\sum_{t=0}^{T}\mathbb{E}[\|\nabla F(\mathbf{x}_t)\|^2] \leq 8\sqrt{\frac{M_\kappa L(F(\mathbf{x}_0) - F^\star)d\sigma_{\mathrm{DP}}^2}{T}} = \mathcal{O}\left(\sqrt{\frac{d}{T}}\right). \quad (9)$$

**Convergence improvement:** The above result implies that the *order* of the number of iterations $T$ needed for convergence of Algorithm 3 is the same as of DPSGD (Ghadimi & Lan, 2013; Zhang et al., 2017). However, Algorithm 3 has a *constant factor* improvement in the upper bound of its iteration complexity over DPSGD. This improvement results from the presence of $M_\kappa \leq 1$ in the numerator in the RHS of (9). More specifically, if $M_\kappa < 1$, i.e., $\|\nabla F(\mathbf{x}_0)\|^2 < 2L(F(\mathbf{x}_0) - F^\star)$, then the convergence bound reduces by a factor of $\sqrt{1/M_\kappa}$, and DiSK has clear theoretical improvement over vanilla DPSGD. **Case I:** For ($\mu$-strongly) convex problems, it is guaranteed that $2\mu(F(\mathbf{x}_0) - F^\star) \leq \|\nabla F(\mathbf{x}_0)\|^2 \leq 2L(F(\mathbf{x}_0) - F^\star)$. Therefore, the factor $M_\kappa \in [\mu/L, 1]$. **Case II:** When training highly non-convex deep learning models, the Lipschitz constant $L$ can be large (Herrera et al., 2020), and $2L(F(\mathbf{x}_0) - F^\star)$ can be much larger than $\|\nabla F(\mathbf{x}_0)\|^2$, which results in a considerable algorithm performance improvement compared to vanilla DPSGD.

While the above corollary is for the case of $\gamma = -1$, we can obtain performance improvement for $\gamma \neq -1$ as well. The choice $\gamma = -1$ is optimized for the *worst case* (function satisfying our assumptions) based on our upper-bound in (8). However, it is possible that for functions satisfying additional assumptions, other choices of $\gamma$ lead to better convergence results. This fact is further explained after presenting the proof of the theorem in Appendix B.2.

As the last remark in this subsection, notice that our iteration complexity improvement does not require any additional assumption on the problem. In contrast, existing works with convergence improvement require additional assumptions, e.g., on the correlation between the gradients (Zhang et al., 2024a), or on the trace of Hessian (Choquette-Choo et al., 2024; Li et al., 2022).

## 4.2 PRIVACY-UTILITY TRADE-OFF

To provide DP, the Subsampled Gaussian mechanism is used in Lines 4 and 5 of Algorithm 3. Instead of $\nabla f(\mathbf{x}_t; \xi)$, we treat $\frac{1-\kappa}{\kappa\gamma}\nabla f(\mathbf{x}_t - \gamma\mathbf{d}_{t-1}; \xi) + (1 - \frac{1-\kappa}{\kappa\gamma})\nabla f(\mathbf{x}_t; \xi)$ as the per-sample gradient and apply the Subsampled Gaussian mechanism to privatize it. Therefore, Algorithm 3 and DPSGD share a similar privacy guarantee, as the privacy proof directly follows the Subsampled Gaussian mechanism and the composition of $T$ iterations in Theorem 1. Specifically, by combining Theorem 1 and Theorem 2, we obtain the following privacy-utility trade-off:

**Theorem 3** *Assume A1-A3 holds. With sufficiently large $N$, run Algorithm 3 for $T = \frac{\sqrt{2}N\epsilon}{C\sqrt{d\ln(1/\delta)}}$ iterations, and choose $\kappa, \beta$ according to the choices in Corollary 1, then*

$$\frac{1}{T}\sum_{t=0}^{T} \mathbb{E}\left\|\nabla F(\mathbf{x}_t)\right\|^2 \leq \frac{4C\sqrt{2M_\kappa L(F(\mathbf{x}_0) - F^\star)d\ln(1/\delta)}}{N\epsilon} = \mathcal{O}\left(\frac{\sqrt{d\ln(1/\delta)}}{N\epsilon}\right) \quad (10)$$

Similar to our convergence result, compared to vanilla DPSGD, the privacy-utility trade-off of Algorithm 3 reduces by a constant factor of $\sqrt{1/M_\kappa}$. To the best of our knowledge, this is the first theoretical result on the utility improvement of a DPSGD-type algorithm without any additional assumptions on the problem.

## 5 NUMERICAL EXPERIMENTS

We perform extensive pre-training and fine-tuning experiments on various image classification (CV) and natural language processing (NLP) tasks using different base algorithms, privacy budgets, and models. The implementation details of the experiments are given in Appendix C.1. An implementation of the algorithm is in `https://github.com/pytorch/opacus/tree/main/research/disk_optimizer`.

## 5.1 EXPERIMENT SETTINGS

**Dataset:** We train the models on one synthetic dataset, four CV datasets, including MNIST (Deng, 2012), CIFAR-10/CIFAR-100 (Krizhevsky et al., 2009), and ImageNet-1k (Deng et al., 2009), and three NLP dataset, including GLUE (Wang et al., 2018), E2E (Novikova et al., 2017), and DART (Nan et al., 2021).

**Model:** For the CV tasks, we use three different models, including a 5-layer CNN, WideResNet (WRN) (De et al., 2022), and ViT (Dosovitskiy et al., 2020), representing three typical CV model structures. For the NLP task, we use the RoBERTa (Liu et al., 2019) and the GPT-2 (Radford et al., 2019) models. For pre-training, the models are initialized with random weights. In fine-tuning ViT, RoBERTa, and GPT-2, we directly use the checkpoints on HuggingFace (Wolf et al., 2020).

**Algorithm:** We use the DP version of SGD and Adam for CV tasks, and AdamW for NLP tasks as baselines. Then, we apply DiSK to compare their performance. Additional results on LoRA (Li et al., 2021) are given in Appendix C.4. In the results, we use **KF-** to denote the privatized version of the algorithms with DiSK. We use sample without replacement with fixed batch size.

**Hyper-parameter choices:** We tune the hyper-parameters using a grid search. Specifically, we conduct a grid search on the batch size $B$, total epochs $E = NT/B$, and step size $\eta$ for each given privacy budget $\epsilon$. For all experiments, we fix the privacy parameter $\delta = 1/N^{1.1}$ to obtain a reasonable privacy notion. Detailed hyper-parameter choices are discussed in Appendix C.2.

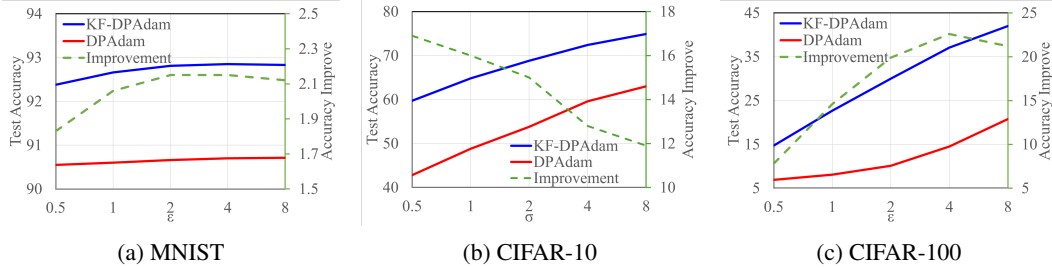

Figure 1: Test accuracy of training from scratch on MNIST, CIFAR-10, and CIFAR-100 datasets with and without DiSK for different privacy $\epsilon$. The green lines show the improvement of DiSK.

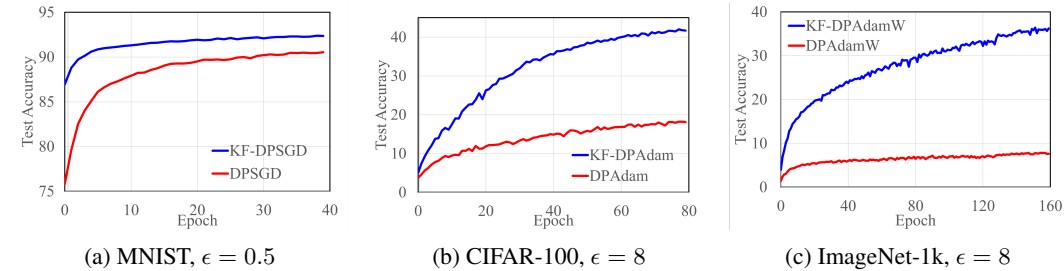

Figure 2: Test accuracy on MNIST, CIFAR-100, and validation accuracy on ImageNet-1k datasets training from scratch with and without DiSK for fixed privacy budgets.

Table 1: Test accuracy of fine-tuning result on the GLUE dataset.

| Task | ($\epsilon = \infty$) Non-DP | $\epsilon = 6.7$ | | | $\epsilon = 1$ | | |
|------|------|------|------|------|------|------|------|
| | | DP | **KF-DP** | **KF-DPLora** | DP | **KF-DP** | **KF-DPLora** |
| MNLI | 87.6 | 83.2 | 84.8 | 85.9 | 80.7 | 82.0 | 84.7 |
| QNLI | 92.8 | 87.5 | 88.9 | 90.5 | 86.0 | 88.7 | 90.3 |
| SST-2 | 94.8 | 91.5 | 92.8 | 93.1 | 91.4 | 91.5 | 92.9 |
| QQP | 91.9 | 85.8 | 88.5 | 89.0 | 84.2 | 86.9 | 87.8 |

## 5.2 NUMERICAL RESULTS

**CV tasks:** We first train the CV models with randomly initialized weights on different image datasets. The results for 5-layer CNN on the MNIST dataset, 5-layer CNN on the CIFAR-10 dataset, and WRN-16-4 on the CIFAR-100 datasets with different privacy budgets are given in Figure 1. DiSK significantly outperforms the base algorithm across all used privacy budgets.

The test accuracy curves during the training for 5-layer CNN on the MNIST dataset, WRN-16-4 on the CIFAR-100 dataset, and ViT-small on the ImageNet-1k dataset are given in Figure 2. The optimizer with DiSK converges faster than the base algorithm on all tasks and reaches a higher final accuracy at a given privacy budget. The test accuracy of CIFAR-100 achieves $41.8\%$, and ImageNet-1k achieves $36.89\%$, which outperforms the SOTA results that apply data augmentation under the same privacy budget ($40.6\%$ for CIFAR-100 (Bao et al., 2024) and $33.56\%$ for ImageNet-1k (De et al., 2022)). Additional comparisons on different models, algorithms, and fine-tuning CIFAR-100 are given in Appendix C.3.

**NLP tasks:** We fine-tune a pre-trained RoBERTa-base model on the GLUE datasets. The final test accuracy for is given in Table 1. We follow the same training script and hyper-parameter choices in the experiments as Bu et al. (2024). Compared with the base algorithm (DPAdamW), DiSK improves the final accuracy on all tasks by at least $3.8\%$ when $\epsilon = 1$ and $1.3\%$ when $\epsilon = 6.7$. Additional results for text generation tasks on the E2E ad DART datasets are given in Appendix C.4.

**Ablation study:** We conduct ablation studies on the choice of the hyper-parameters of DiSK, specifically, how $\kappa, \gamma$ impact the algorithm performance. The results are presented in Appendix C.3.

**Improvements over the SOTA:** Table 6 in Appendix C.5 summarizes the improvements of DiSK over the SOTA on different tasks.

## ACKNOWLEDGEMENT

X. Zhang and M. Razaviyayn's work is supported by a gift from Meta, a gift from Google, and a gift from Amazon. M. Hong acknowledges the Minnesota Supercomputing Institute (MSI) at the University of Minnesota for providing resources that contributed to the research results reported within this paper. URL: `http://www.msi.umn.edu`. M. Hong's work is partially supported by NSF grants CIF-2414372, ECCS-2426064, and NSF EPCN-2311007.

## REPRODUCIBILITY STATEMENT

For the algorithm implementation, we provide a link to an anonymous downloadable source code in Section 5. We also discussed the data, model, and algorithms used in the experiment in Section 5 and in Appendix C. For our theoretical results provided in Section 4, we also include clear explanations of the assumptions in the section. The complete proof of the theorems can be found in Appendix B.

## BROADER IMPACT

This paper presents work that aims to advance the field of Machine Learning, combining optimization with signal processing and control societies. There are many potential societal consequences of our work, none of which we feel must be specifically highlighted here.

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

## A  ADDITIONAL DISCUSSION

### A.1  BACKGROUND ON KALMAN FILTER

In this section, we provide an introduction and derivation of the Kalman filter. Kalman filter is introduced in (Kalman, 1960) and widely used for control systems in accurately estimating system states with noisy observation and known system dynamics. Kalman filter assumes the system is a linear (time-invariant) system:

$$\boldsymbol{\theta}_t = \mathbf{A}\boldsymbol{\theta}_{t-1} + \mathbf{u}_t + \mathbf{v}_t, \qquad \text{(System update)}$$

$$\boldsymbol{\psi}_t = \mathbf{C}\boldsymbol{\theta}_t + \mathbf{w}_t, \qquad \text{(Observation)}$$

where $\mathbf{A}, \mathbf{C}$ are known matrix, and $\mathbf{v}_t \sim \mathcal{N}(0, \Sigma_{\mathbf{v}}), \mathbf{w}_t \sim \mathcal{N}(, \Sigma_{\mathbf{w}})$ are independent white noise following Gaussian distributions with known covariance matrices $\Sigma_{\mathbf{v}}, \Sigma_{\mathbf{w}}$, and $\mathbf{u}_t$ is known input signal. The goal of the Kalman filter is to estimate $\boldsymbol{\theta}_t$ with the observation of $\boldsymbol{\psi}_t$ with the least mean-square error and serves as the Best Linear Unbiased Estimator (BLUE) (Welch et al., 1995).

**Derivation:** First, we denote the estimation of $\boldsymbol{\theta}_t$ as $\tilde{\boldsymbol{\theta}}_t$, and the covariance of $\boldsymbol{\theta}_t - \tilde{\boldsymbol{\theta}}_t$ as

$$\mathbf{P}_t = \mathbb{E}[(\boldsymbol{\theta}_t - \tilde{\boldsymbol{\theta}}_t)(\boldsymbol{\theta}_t - \tilde{\boldsymbol{\theta}}_t)^\top].$$

Then, based on the knowledge at time $t-1$, the system dynamics give an unbiased prediction:

$$\tilde{\boldsymbol{\theta}}_{t|t-1} = \mathbf{A}\tilde{\boldsymbol{\theta}}_{t-1} + \mathbf{u}_t, \qquad (11)$$

and its covariance is

$$\mathbf{P}_{t|t-1} = \mathbb{E}[(\boldsymbol{\theta}_t - \tilde{\boldsymbol{\theta}}_{t|t-1})(\boldsymbol{\theta}_t - \tilde{\boldsymbol{\theta}}_{t|t-1})^\top] = \mathbf{A}\mathbf{P}_{t-1}\mathbf{A}^\top + \Sigma_{\mathbf{v}}. \qquad (12)$$

With $\tilde{\boldsymbol{\theta}}_{t|t-1}$, we have an (unbiased) prediction of the observation $\tilde{\boldsymbol{\psi}}_{t|t-1} = \mathbf{C}\tilde{\boldsymbol{\theta}}_{t|t-1}$ at time $t-1$. At time $t$, by observing $\boldsymbol{\psi}_t$, we have the prediction error $\Delta\boldsymbol{\psi}_t = \boldsymbol{\psi}_t - \tilde{\boldsymbol{\psi}}_{t|t-1} = \boldsymbol{\psi}_t - \mathbf{C}\tilde{\boldsymbol{\theta}}_{t|t-1}$.

Since the system is linear, we would like to use the prediction error to correct the prediction:

$$\tilde{\boldsymbol{\theta}}_t = \tilde{\boldsymbol{\theta}}_{t|t-1} + \mathbf{K}_t \Delta \boldsymbol{\psi}_t = \mathbf{K}_t \boldsymbol{\psi}_t + (\mathbf{I} - \mathbf{K}_t \mathbf{C}) \tilde{\boldsymbol{\theta}}_{t|t-1}. \tag{13}$$

The goal of the Kalman filter is to minimize the mean-square error: $\min_{\mathbf{K}} \mathbb{E}[\left\| \boldsymbol{\theta}_t - \tilde{\boldsymbol{\theta}}_t \right\|^2]$, which is equivalent to minimizing $\mathrm{tr}(\mathbf{P}_t)$. From the definition of $\tilde{\boldsymbol{\theta}}_t$, we have:

$$\begin{aligned}
\mathbf{P}_t &= \mathbb{E}[(\boldsymbol{\theta}_t - \tilde{\boldsymbol{\theta}}_t)(\boldsymbol{\theta}_t - \tilde{\boldsymbol{\theta}}_t)^\top] \\
&= \mathbb{E}[(\boldsymbol{\theta}_t - \mathbf{K}_t \boldsymbol{\psi}_t + (\mathbf{I} - \mathbf{K}_t \mathbf{C})\tilde{\boldsymbol{\theta}}_{t|t-1})(\boldsymbol{\theta}_t - \mathbf{K}_t \boldsymbol{\psi}_t + (\mathbf{I} - \mathbf{K}_t \mathbf{C})\tilde{\boldsymbol{\theta}}_{t|t-1})^\top] \\
&= \mathbb{E}[(\boldsymbol{\theta}_t - \tilde{\boldsymbol{\theta}}_{t|t-1} - \mathbf{K}_t\mathbf{C}(\boldsymbol{\theta}_t - \tilde{\boldsymbol{\theta}}_{t|t-1}) - \mathbf{K}_t\mathbf{w}_t)(\boldsymbol{\theta}_t - \tilde{\boldsymbol{\theta}}_{t|t-1} - \mathbf{K}_t\mathbf{C}(\boldsymbol{\theta}_t - \tilde{\boldsymbol{\theta}}_{t|t-1}) - \mathbf{K}_t\mathbf{w}_t)^\top] \\
&= \mathbf{P}_{t|t-1} - \mathbf{K}_t\mathbf{C}\mathbf{P}_{t|t-1} - (\mathbf{K}_t\mathbf{C}\mathbf{P}_{t|t-1})^\top + \mathbf{K}_t(\mathbf{C}\mathbf{P}_{t|t-1}\mathbf{C}^\top + \Sigma_{\mathbf{w}})\mathbf{K}_t^\top.
\end{aligned}$$

Taking partial derivative to the trace of $\mathbf{P}_t$ with respect to $\mathbf{K}_t$, and set it to zero, we have:

$$\frac{\partial \mathrm{tr}(\mathbf{P}_t)}{\partial \mathbf{K}_t} = -2(\mathbf{C}\mathbf{P}_{t|t-1})^\top + 2(\mathbf{C}\mathbf{P}_{t|t-1}\mathbf{C}^\top + \Sigma_{\mathbf{w}})\mathbf{K}_t^\top = 0,$$

which gives

$$\mathbf{K}_t = \mathbf{P}_{t|t-1}\mathbf{C}^\top(\mathbf{C}\mathbf{P}_{t|t-1}\mathbf{C}^\top + \Sigma_{\mathbf{w}})^{-1}. \tag{14}$$

Substitute $\mathbf{K}_t$ back to $\mathbf{P}_t$, we can simplify

$$\begin{aligned}
\mathbf{P}_t &= \mathbf{P}_{t|t-1} - \mathbf{K}_t\mathbf{C}\mathbf{P}_{t|t-1} - (\mathbf{K}_t\mathbf{C}\mathbf{P}_{t|t-1})^\top + \mathbf{K}_t(\mathbf{C}\mathbf{P}_{t|t-1}\mathbf{C}^\top + \Sigma_{\mathbf{w}})\mathbf{K}_t^\top \\
&= \mathbf{P}_{t|t-1} - \mathbf{K}_t\mathbf{C}\mathbf{P}_{t|t-1} = (\mathbf{I} - \mathbf{K}_t\mathbf{C})\mathbf{P}_{t|t-1}.
\end{aligned} \tag{15}$$

Combining (11)-(15) together, we have the update of the Kalman filter:

$$\begin{aligned}
\tilde{\boldsymbol{\theta}}_{t|t-1} &= \mathbf{A}\tilde{\boldsymbol{\theta}}_{t-1} + \mathbf{u}_t \\
\mathbf{P}_{t|t-1} &= \mathbf{A}\mathbf{P}_{t-1}\mathbf{A}^\top + \Sigma_{\mathbf{v}} \\
\mathbf{K}_t &= \mathbf{P}_{t|t-1}\mathbf{C}^\top(\mathbf{C}\mathbf{P}_{t|t-1}\mathbf{C}^\top + \Sigma_{\mathbf{w}})^{-1} \\
\tilde{\boldsymbol{\theta}}_t &= (\mathbf{I} - \mathbf{K}_t\mathbf{C})\tilde{\boldsymbol{\theta}}_{t|t-1} + \mathbf{K}_t\boldsymbol{\psi}_t \\
\mathbf{P}_t &= (\mathbf{I} - \mathbf{K}_t\mathbf{C})\mathbf{P}_{t|t-1}.
\end{aligned}$$

**Variants of Kalman filter:** Kalman filter is designed for estimating the states following linear dynamics and achieves optimal performance, i.e., gives the smallest mean square error of the estimation when the system is linear (Welch et al., 1995). Extended Kalman filter (EKF) and unscented Kalman filter (UKF) are developed to deal with non-linear systems. EKF linearizes the non-linear system at each step and performs the Kalman filter on the linearized system (Ribeiro, 2004), while UKF takes the effect of the system non-linearity to the noise distribution into consideration and applies the unscented transform on the noise distribution and applies the Kalman filter on the system and the noise distribution (Wan & Van Der Merwe, 2001). Other extensions of the Kalman filter have been developed for special cases, including multiplicative noise and noisy input $\mathbf{u}_t$ (Wu et al., 2016).

## A.2 OTHER SYSTEM DYNAMICS FOR DPSGD

In this section, we would like to discuss other possible formulations of the system dynamics for DPSGD to apply the Kalman filter.

**Optimization variable and gradient version 1:** Other than only using the gradients' dynamics in the main paper, we can construct the dynamic system with both the optimization variable $\mathbf{x}$ and the gradient $\nabla F(\mathbf{x})$ as its states (Vuckovic, 2018):

$$\begin{bmatrix} \mathbf{x}_{t+1} \\ \nabla F(\mathbf{x}_{t+1}) \end{bmatrix} = \begin{bmatrix} \mathbf{I} & -\eta\mathbf{I} \\ 0 & \mathbf{I} \end{bmatrix} \begin{bmatrix} \mathbf{x}_t \\ \nabla F(\mathbf{x}_t) \end{bmatrix} + \begin{bmatrix} \eta\mathbf{w}_t \\ 0 \end{bmatrix},$$
$$\mathbf{y}_t = \mathbf{x}_t.$$

However, this dynamic is inaccurate as the dynamics of the gradient are simplified to $\nabla F(\mathbf{x}_{t+1}) = \nabla F(\mathbf{x}_t)$. Although the update can further incorporate with the momentum methods, i.e., by adding a momentum $\mathbf{m}_t$ into the system, it fails to reveal the actual dynamic of the system.

**Optimization variable and gradient version 2:** Instead of treating the gradient as a constant, we can assume the Hessian $\mathbf{H}$ is a constant and utilize the Hessian to reveal the gradient dynamics:

$$\begin{bmatrix} \mathbf{x}_{t+1} \\ \mathbf{x}_t \\ \nabla F(\mathbf{x}_t) \end{bmatrix} = \begin{bmatrix} \mathbf{I} - \eta\mathbf{H} & \eta\mathbf{H} & -\eta\mathbf{I} \\ \mathbf{I} & 0 & 0 \\ \mathbf{H} & -\mathbf{H} & \mathbf{I} \end{bmatrix} \begin{bmatrix} \mathbf{x}_t \\ \mathbf{x}_{t-1} \\ \nabla F(\mathbf{x}_{t-1}) \end{bmatrix} + \begin{bmatrix} \eta\mathbf{w}_t \\ 0 \\ 0 \end{bmatrix},$$

$$\mathbf{y}_t = \mathbf{x}_t.$$

This system is more accurate in evaluating the gradient at the cost of using an extra $\mathbf{x}_{t-1}$ state and a larger transition matrix. For non-linear problems $F(\mathbf{x})$, where $\mathbf{H}$ is not a constant, we can apply the extended Kalman filter and replace $\mathbf{H}$ with $\mathbf{H}_t$ that linearizes the problem at each step $t$.

**Gradient and Hessian:** In work (Bittner & Pronzato, 2004), the dynamics of the gradient and Hessian have been used to construct the system:

$$\begin{bmatrix} \nabla F(\mathbf{x}_{t+1}) \\ \mathbf{h}_{t+1} \end{bmatrix} = \begin{bmatrix} \mathbf{I} & \Delta\mathbf{X}_t \\ 0 & \mathbf{I} \end{bmatrix} \begin{bmatrix} \nabla F(\mathbf{x}_t) \\ \mathbf{h}_t \end{bmatrix},$$

$$\mathbf{g}_t = \nabla F(\mathbf{x}_t) + \mathbf{w}_t,$$

where $\mathbf{h}_t = [\mathbf{H}_{1,1}, \ldots, \mathbf{H}_{i,i+j}, \ldots, \mathbf{H}_{d,d}]^\top$, with $j \in [0, \ldots, d-i]$ represents the entries in the upper triangular part of the Hessian matrix. $\Delta\mathbf{X}_t$ is constructed such that $\Delta\mathbf{X}_t\mathbf{h}_t = \mathbf{H}_t(\mathbf{x}_{t+1} - \mathbf{x}_t)$. The system treats the Hessian matrix as a constant matrix (i.e., $\mathbf{h}_{t+1} = \mathbf{h}_t$), and the transition matrix is of size $(d + d(d-1)/2) \times d + d(d-1)/2$.

Although there are different ways to construct the Kamlan filter for gradient noise reduction, the above systems are not implementable in practical deep-learning applications. Because the transition matrices of these systems are non-diagonal, the Kalman filters have non-diagonal gain $\mathbf{K}_t$ and $\mathbf{P}_t$. Therefore, the matrix inversion operation is unavoidable when Kalman filters are implemented based on these systems. The computation complexity for the matrix inversion can be $\mathcal{O}(d^3)$ to $\mathcal{O}(d^6)$, and the memory consumption is $\mathcal{O}(d^2)$ to $\mathcal{O}(d^4)$ for storing the matrices of the Kalman filter.

### A.3 ALGORITHM SIMPLIFICATION

In this section, we explain how the simplification proceeds from Algorithm 2 to Algorithm 3. Recall that updates of Algorithm 2 is

$$\tilde{\mathbf{g}}_{t|t-1} = \tilde{\mathbf{g}}_{t-1} + \tilde{\mathbf{H}}_t(\mathbf{x}_t - \mathbf{x}_{t-1}) \qquad \textit{(Prediction)}$$

$$\mathbf{P}_{t|t-1} = \mathbf{P}_{t-1} + \Sigma_\mathbf{H} + \Sigma_\mathbf{v}$$

$$\mathbf{K}_t = \mathbf{P}_{t|t-1}\,\mathbb{E}[\mathbf{C}_t]^\top \left(\Sigma_\mathbf{w} + \mathbb{E}[\mathbf{C}_t]\left(\Sigma_\mathbf{C}\mathbf{S}_t + \mathbf{P}_{t|t-1}\right)\mathbb{E}[\mathbf{C}_t]^\top - \Sigma_\mathbf{H}\right)^{-1}$$

$$\tilde{\mathbf{g}}_t = \tilde{\mathbf{g}}_{t|t-1} + \mathbf{K}_t(\mathbf{g}_t - \mathbb{E}[\mathbf{C}_t]\tilde{\mathbf{g}}_{t|t-1}) \qquad \textit{(Correction)}$$

$$\mathbf{P}_t = (\mathbf{I} - \mathbf{K}_t\,\mathbb{E}[\mathbf{C}_t])\mathbf{P}_{t|t-1}$$

$$\mathbf{S}_t = \mathbb{E}[\tilde{\mathbf{g}}_t\tilde{\mathbf{g}}_t^\top],$$

where $\mathbf{g}_t = \frac{1}{B}\sum_{\xi\in\mathcal{B}_t}\text{clip}\left(\nabla f(\mathbf{x}_t; \xi), C\right) + \mathbf{w}_t$. We apply four steps of simplification, including

1. Replacing random $\mathbf{C}_t$ with constant $\mathbf{I}_d$.

2. Use finite difference to estimate $\mathbf{H}_t\mathbf{d}_{t-1}$.

3. Replace $\Sigma_\mathbf{H}$ with diagonal matrix $\sigma_H^2\mathbf{I}_d$, $\Sigma_\mathbf{v}$ with $\sigma_\mathbf{v}^2\mathbf{I}_d$, and $\Sigma_\mathbf{w}$ with $\sigma_\mathbf{w}^2\mathbf{I}_d$.

4. Use fixed filter gain $\kappa\mathbf{I}$.

**Step 1.** By replacing $\mathbf{C}_t$ with $\mathbf{I}_d$, and $\Sigma_\mathbf{C} = 0$, the update becomes

$$\tilde{\mathbf{g}}_{t|t-1} = \tilde{\mathbf{g}}_{t-1} + \tilde{\mathbf{H}}_t \cdot (\mathbf{x}_t - \mathbf{x}_{t-1}) \qquad \textit{(Prediction)}$$

$$\mathbf{P}_{t|t-1} = \mathbf{P}_{t-1} + \Sigma_\mathbf{H} + \Sigma_\mathbf{v}$$

$$\mathbf{K}_t = \mathbf{P}_{t|t-1}\left(\Sigma_\mathbf{w} + \mathbf{P}_{t|t-1} - \Sigma_\mathbf{H}\right)^{-1}$$

$$\tilde{\mathbf{g}}_t = \tilde{\mathbf{g}}_{t|t-1} + \mathbf{K}_t(\mathbf{g}_t - \tilde{\mathbf{g}}_{t|t-1}) \qquad \textit{(Correction)}$$

$$\mathbf{P}_t = (\mathbf{I} - \mathbf{K}_t)\mathbf{P}_{t|t-1}.$$

**Step 2.** By using the finite difference to estimate $\mathbf{H}_t\mathbf{d}_{t-1}$, the prediction step becomes:

$$\tilde{\mathbf{g}}_{t|t-1} = \tilde{\mathbf{g}}_{t-1} + \frac{1}{B}\sum_{\xi\in\mathcal{B}_t}\frac{\nabla f(\mathbf{x}_t + \gamma\mathbf{d}_{t-1}; \xi) - \nabla f(\mathbf{x}_t; \xi)}{\gamma} \qquad \textit{(Prediction)}.$$

**Step 3.** By replacing $\Sigma$'s with diagonal matrix $\sigma^2\mathbf{I}_d$'s, the update becomes:

$$\tilde{\mathbf{g}}_{t|t-1} = \tilde{\mathbf{g}}_{t-1} + \frac{1}{B}\sum_{\xi\in\mathcal{B}_t}\frac{\nabla f(\mathbf{x}_t + \gamma\mathbf{d}_{t-1}; \xi) - \nabla f(\mathbf{x}_t; \xi)}{\gamma} \qquad \textit{(Prediction)}$$

$$p_{t|t-1} = p_{t-1} + \sigma_H^2 + \sigma_{\mathbf{v}}^2, \mathbf{P}_{t|t-1} = p_{t|t-1}\mathbf{I}_d$$

$$k_t = \frac{p_{t|t-1}}{p_{t|t-1} + \sigma_{\mathbf{w}}^2 - \sigma_H^2} = \frac{p_{t-1} + \sigma_H^2 + \sigma_{\mathbf{v}}^2}{p_{t-1} + \sigma_{\mathbf{w}}^2 + \sigma_{\mathbf{v}}^2}, \mathbf{K}_t = k_t\mathbf{I}_d$$

$$\tilde{\mathbf{g}}_t = \tilde{\mathbf{g}}_{t|t-1} + k_t(\mathbf{g}_t - \tilde{\mathbf{g}}_{t|t-1}) \qquad \text{(Correction)}$$

$$p_t = (1-k_t)p_{t|t-1} = \frac{(\sigma_{\mathbf{w}}^2 - \sigma_H^2)(p_{t-1} + \sigma_H^2 + \sigma_{\mathbf{v}}^2)}{p_{t-1} + \sigma_{\mathbf{w}}^2 + \sigma_{\mathbf{v}}^2}.$$

**Step 4.** As discussed in the main paper, the update of $k_t, p_t$ becomes:

$$k_t = \frac{p_{t-1} + \sigma_H^2 + \sigma_{\mathbf{v}}^2}{p_{t-1} + \sigma_{\mathbf{w}}^2 + \sigma_{\mathbf{v}}^2},$$

$$p_t = \frac{(\sigma_{\mathbf{w}}^2 - \sigma_H^2)(p_{t-1} + \sigma_H^2 + \sigma_{\mathbf{v}}^2)}{p_{t-1} + \sigma_{\mathbf{w}}^2 + \sigma_{\mathbf{v}}^2}.$$

Therefore, solving the recurrence relation of the sequence $\{p_t\}$, we have $p_t$ converges to $p_\infty = \frac{\sqrt{\sigma_H^2 + \sigma_{\mathbf{v}}^2}\sqrt{4\sigma_{\mathbf{w}}^2 - 3\sigma_H^2 + \sigma_{\mathbf{v}}^2} - (\sigma_H^2 + \sigma_{\mathbf{v}}^2)}{2}$, $k_t$ converges to $k_\infty = \frac{p_\infty + \sigma_H^2 + \sigma_{\mathbf{v}}^2}{p_\infty + \sigma_{\mathbf{w}}^2 + \sigma_{\mathbf{v}}^2}$, with rate $c_k = \frac{2\sigma_{\mathbf{w}}^2 + 3\sigma_H^2 + \sigma_{\mathbf{v}}^2 - \sqrt{(\sigma_{\mathbf{v}}^2 + \sigma_H^2)(4\sigma_{\mathbf{w}}^2 + \sigma_{\mathbf{v}}^2 - 3\sigma_H^2)}}{2\sigma_{\mathbf{w}}^2 + 3\sigma_H^2 + \sigma_{\mathbf{v}}^2 + \sqrt{(\sigma_{\mathbf{v}}^2 + \sigma_H^2)(4\sigma_{\mathbf{w}}^2 + \sigma_{\mathbf{v}}^2 - 3\sigma_H^2)}}$.

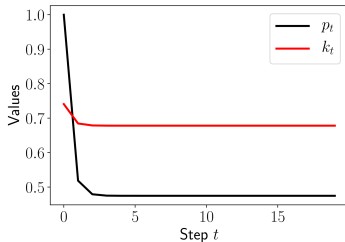

Figure 3: Convergence of $k_t, p_t$.

Therefore, we define $\kappa = k_\infty$ and replace $k_t$ with $\kappa$, and the update becomes:

$$\tilde{\mathbf{g}}_{t|t-1} = \tilde{\mathbf{g}}_{t-1} + \frac{1}{B}\sum_{\xi\in\mathcal{B}_t}\frac{\nabla f(\mathbf{x}_t + \gamma\mathbf{d}_{t-1}; \xi) - \nabla f(\mathbf{x}_t; \xi)}{\gamma} \qquad \text{(Prediction)}$$

$$\tilde{\mathbf{g}}_t = \tilde{\mathbf{g}}_{t|t-1} + \kappa(\mathbf{g}_t - \tilde{\mathbf{g}}_{t|t-1}) \qquad \text{(Correction)}$$

Rearrange the terms, we have:

$$\tilde{\mathbf{g}}_t = (1-\kappa)\tilde{\mathbf{g}}_t + \kappa\mathbf{g}_t + (1-\kappa)\frac{1}{B}\sum_{\xi\in\mathcal{B}_t}\frac{\nabla f(\mathbf{x}_t + \gamma\mathbf{d}_{t-1}; \xi) - \nabla f(\mathbf{x}_t; \xi)}{\gamma}$$

$$= (1-\kappa)\tilde{\mathbf{g}}_t + \kappa\hat{\mathbf{g}}_t, \text{ with}$$

$$\hat{\mathbf{g}}_t = \mathbf{g}_t + \frac{1-\kappa}{\kappa}\frac{1}{B}\sum_{\xi\in\mathcal{B}_t}\frac{\nabla f(\mathbf{x}_t + \gamma\mathbf{d}_{t-1}; \xi) - \nabla f(\mathbf{x}_t; \xi)}{\gamma}$$

$$= \frac{1}{B}\sum_{\xi\in\mathcal{B}_t}\left(\frac{1-\kappa}{\kappa\gamma}\nabla f(\mathbf{x}_t + \gamma\mathbf{d}_{t-1}; \xi) + \left(1 - \frac{1-\kappa}{\kappa\gamma}\right)\nabla f(\mathbf{x}_t; \xi)\right) + \mathbf{w}_t,$$

where we substitute $\mathbf{g}_t = \frac{1}{B}\sum_{\xi\in\mathcal{B}_t}\text{clip}\left(\nabla f(\mathbf{x}_t; \xi), C\right) + \mathbf{w}_t$. Then, by privatizing $\hat{\mathbf{g}}_t$ and rename it as $\mathbf{g}_t$, we have:

$$\mathbf{g}_t = \frac{1}{B}\sum_{\xi\in\mathcal{B}_t}\text{clip}\left(\frac{1-\kappa}{\kappa\gamma}\nabla f(\mathbf{x}_t + \gamma\mathbf{d}_{t-1}; \xi) + \left(1 - \frac{1-\kappa}{\kappa\gamma}\right)\nabla f(\mathbf{x}_t; \xi), C\right) + \mathbf{w}_t$$

$$\tilde{\mathbf{g}}_t = (1-\kappa)\tilde{\mathbf{g}}_t + \kappa\mathbf{g}_t,$$

which is the update of Algorithm 3 (Lines 5 and 6).

### A.4 CONNECTION BETWEEN DISK AND NAG AND STORM

Algorithm 3 has an implicit connection to the (unified) Nesterov Accelerated Gradient (NAG) method (Shen et al., 2023; Sutskever et al., 2013) and the Stochastic Recursive Momentum

(STORM) algorithm (Cutkosky & Orabona, 2019). The update of (half-shifted) NAG can be written as (Sutskever et al., 2013):

$$\mathbf{m}_t = \mu\mathbf{m}_{t-1} + \eta\nabla F(\mathbf{x}_t - \mu\mathbf{m}_{t-1})$$
$$\mathbf{x}_{t+1} = \mathbf{x}_t - \mathbf{m}_t, \tag{16}$$

and the update of STORM writes (Cutkosky & Orabona, 2019):

$$\mathbf{m}_t = (1-\alpha)\mathbf{m}_{t-1} + \alpha\nabla f(\mathbf{x}_t, \xi_t) + (1-\alpha)(\nabla f(\mathbf{x}_t, \xi_t) - \nabla f(\mathbf{x}_{t-1}, \xi_t))$$
$$\mathbf{x}_{t+1} = \mathbf{x}_t - \eta\mathbf{m}_t, \tag{17}$$

In comparison, in Algorithm 3, by letting the update of OptimizerUpdate be SGD, and assuming clipping is inactive, $\mathbf{w}_t = 0$, i.e., without DP, $\epsilon \to \infty$, the DiSK becomes:

$$\tilde{\mathbf{g}}_t = (1-\kappa)\tilde{\mathbf{g}}_{t-1} + \frac{\kappa}{B}\sum_{\xi\in\mathcal{B}_t}\left(\frac{1-\kappa}{\kappa\gamma}\nabla f(\mathbf{x}_t + \gamma\mathbf{d}_{t-1}; \xi) + \left(1 - \frac{1-\kappa}{\kappa\gamma}\right)\nabla f(\mathbf{x}_t; \xi)\right)$$

$$= (1-\kappa)\tilde{\mathbf{g}}_{t-1} + \frac{\kappa}{B}\sum_{\xi\in\mathcal{B}_t}\nabla f(\mathbf{x}_t; \xi) + \frac{1-\kappa}{B\gamma}\sum_{\xi\in\mathcal{B}_t}(\nabla f(\mathbf{x}_t + \gamma\mathbf{d}_{t-1}; \xi) - \nabla f(\mathbf{x}_t; \xi)) \tag{18}$$

$$\mathbf{x}_{t+1} = \mathbf{x}_t - \eta\tilde{\mathbf{g}}_t.$$

Comparing the above updates, we observe that (16) and (17) are two special cases of (18) with specific choices of the parameters. Specifically, by letting $\gamma = \frac{1-\kappa}{\kappa} > 0, \eta = \kappa, 1-\kappa = \mu, B = N$, we have $1 - \frac{1-\kappa}{\kappa\gamma} = 0$, and the update of $\tilde{\mathbf{g}}_t$ becomes $\tilde{\mathbf{g}}_t = \mu\tilde{\mathbf{g}}_{t-1} + \eta\nabla F(\mathbf{x}_t - \frac{\mu}{\eta}\tilde{\mathbf{g}}_{t-1})$, and DiSK becomes NAG; on the other hand, by letting $\gamma = -1, \kappa = \alpha, B = 1$, we have $\mathbf{x}_t + \gamma\mathbf{d}_{t-1} = \mathbf{x}_t - (\mathbf{x}_t - \mathbf{x}_{t-1}) = \mathbf{x}_{t-1}$, and the update of $\tilde{\mathbf{g}}_t$ becomes $\tilde{\mathbf{g}}_t = (1-\alpha)\tilde{\mathbf{g}}_{t-1} + \alpha\nabla f(\mathbf{x}_t; \xi_t) + (1-\alpha)(\nabla f(\mathbf{x}_t; \xi) - \nabla f(\mathbf{x}_{t-1}; \xi))$, which matches the update of STORM. From this discussion, we see that NAG and STORM, two algorithms for accelerated gradient and variance reduction, respectively, can be unified by DiSK, an algorithm based on Kalman filtering.

With this discussion, we observe that the key difference between NAG and STORM is the choice of $\gamma$. When $\gamma > 0$, the algorithm is close to NAG, which focuses on "exploring" and "accelerating". When $\gamma < 0$, the algorithm is close to STORM, which focuses on "exploiting" and "reducing noise".

## B  PROOF FOR SECTION 4

In this section, we provide detailed proof for the results in Section 4.

We will use the following inequalities in our proofs:

$$\langle\mathbf{a}, \mathbf{b}\rangle \le \frac{1}{2\alpha}\|\mathbf{a}\|^2 + \frac{\alpha}{2}\|\mathbf{b}\|^2, \tag{19}$$

$$\|\mathbf{a} + \mathbf{b}\|^2 \le (1+\alpha)\|\mathbf{a}\|^2 + (1+1/\alpha)\|\mathbf{b}\|^2. \tag{20}$$

In the following sections, we use $\mathbb{E}_t$ to denote the expectation conditioned on all the information before iteration $t$. To prove Theorem 2, we first provide the following lemma to bound the difference between $\tilde{\mathbf{g}}_t$ defined in Line 6 of Algorithm 3, and $\nabla F(\mathbf{x}_t)$. Let us define $\Delta_t = \nabla F(\mathbf{x}_t) - \tilde{\mathbf{g}}_t$. We have:

**Lemma 1** *Assume A1, A2, and A3 holds and choose $C \ge \left(1 + \frac{2(1-\kappa)}{\kappa}\right)G$, we have:*

$$\mathbb{E}_t\|\Delta_t\|^2 \le (1-\kappa)^2\left(1 + 4\eta^2L^2 + |1+\gamma|\left(\kappa + 2\eta^2L^2M_\gamma\right)\right)\|\Delta_{t-1}\|^2$$
$$+ 2\eta^2L^2(1-\kappa)^2\left(2 + |1+\gamma|M_\gamma\right)\|\nabla F(\mathbf{x}_{t-1})\|^2$$
$$+ \kappa^2\left((2+|1+\gamma|)\frac{\sigma_{SGD}^2}{B} + d\sigma_{DP}^2\right), \tag{21}$$

*where we define $M_\gamma = \left(1 + \frac{4(2+1/\kappa+|1+\gamma|)}{\gamma^2}\right)$.*

### B.1  PROOF OF LEMMA 1

First notice that when choosing $C \ge \left(1 + \frac{2(1-\kappa)}{\kappa}\right)G$, the clipping operation is inactive. By the update of $\tilde{\mathbf{g}}_t$ in Line 6 of Algorithm 3, we have:

$$\mathbb{E}_t\|\Delta_t\|^2$$

$$= \mathbb{E}_t \left\| \nabla F(\mathbf{x}_t) - (1-\kappa)\tilde{\mathbf{g}}_{t-1} - \frac{\kappa}{B} \sum_{\xi \in \mathcal{B}_t} \nabla f(\mathbf{x}_t, \xi) - \kappa \mathbf{w}_t \right.$$

$$\left. - \frac{1-\kappa}{\gamma B} \sum_{\xi \in \mathcal{B}_t} (\nabla f(\mathbf{x}_t + \gamma \mathbf{d}_{t-1}; \xi) - \nabla f(\mathbf{x}_t; \xi)) \right\|^2$$

$$\overset{(a)}{=} \mathbb{E}_t \left\| (1-\kappa)(\nabla F(\mathbf{x}_t) - \nabla F(\mathbf{x}_{t-1})) + (1-\kappa)(\nabla F(\mathbf{x}_{t-1}) - \tilde{\mathbf{g}}_{t-1}) \right.$$

$$+ \kappa \left( \nabla F(\mathbf{x}_t) - \frac{1}{B} \sum_{\xi \in \mathcal{B}_t} \nabla f(\mathbf{x}_t, \xi) - \mathbf{w}_t \right)$$

$$\left. - \frac{1-\kappa}{\gamma B} \sum_{\xi \in \mathcal{B}_t} (\nabla f(\mathbf{x}_t + \gamma \mathbf{d}_{t-1}; \xi) - \nabla f(\mathbf{x}_t; \xi)) \right\|^2$$

$$\overset{(b)}{=} \mathbb{E}_t \left\| (1-\kappa) \underbrace{\left( \nabla F(\mathbf{x}_t) - \nabla F(\mathbf{x}_{t-1}) - \frac{1}{B} \sum_{\xi \in \mathcal{B}_t} (\nabla f(\mathbf{x}_t; \xi) - \nabla f(\mathbf{x}_{t-1}; \xi)) \right)}_{:=D_1} \right.$$

$$+ (1-\kappa)\Delta_{t-1} + \kappa \underbrace{\left( \nabla F(\mathbf{x}_t) - \frac{1}{B} \sum_{\xi \in \mathcal{B}_t} \nabla f(\mathbf{x}_t, \xi) \right)}_{:=D_2} - \kappa \mathbf{w}_t$$

$$\left. - (1-\kappa) \frac{1}{B} \sum_{\xi \in \mathcal{B}_t} \underbrace{\left( \frac{1}{\gamma} \nabla f(\mathbf{x}_t + \gamma \mathbf{d}_{t-1}; \xi) + \nabla f(\mathbf{x}_{t-1}; \xi) - \frac{1+\gamma}{\gamma} \nabla f(\mathbf{x}_t; \xi) \right)}_{:=D_3} \right\|^2$$

$$\overset{(c)}{=} (1-\kappa)^2 \mathbb{E}_t \|D_1\|^2 + (1-\kappa)^2 \|\Delta_{t-1}\|^2 + \kappa^2 \mathbb{E}_t[\|D_2\|^2] + (1-\kappa)^2 \mathbb{E}_t \|D_3\|^2 + \kappa^2 \mathbb{E}[\|\mathbf{w}_t\|^2]$$

$$+ 2(1-\kappa)^2 \langle \mathbb{E}_t[D_1], \Delta_{t-1} \rangle + 2(1-\kappa)\kappa \langle \mathbb{E}_t[D_2], \Delta_{t-1} \rangle - 2(1-\kappa)^2 \langle \mathbb{E}_t[D_3], \Delta_{t-1} \rangle$$

$$+ 2\mathbb{E}_t[\langle (1-\kappa)D_1, \kappa D_2 \rangle] - 2(1-\kappa)^2 \mathbb{E}_t[\langle D_1, D_3 \rangle] - 2\mathbb{E}_t[\langle \kappa D_2, (1-\kappa)D_3 \rangle],$$

$$\overset{(d)}{\leq} (1-\kappa)^2(2 + |1+\gamma|) \mathbb{E}_t \|D_1\|^2 + (1-\kappa)^2(1 + \kappa|1+\gamma|) \|\Delta_{t-1}\|^2 + \kappa^2(2 + |1+\gamma|) \mathbb{E}_t[\|D_2\|^2]$$

$$+ (1-\kappa)^2 \left( 1 + \frac{2}{|1+\gamma|} + \frac{1}{\kappa|1+\gamma|} \right) \mathbb{E}_t \|D_3\|^2 + \kappa^2 d\sigma_{\text{DP}}^2, \tag{22}$$

where $(a)$ we add and subtract $(1-\kappa)\nabla F(\mathbf{x}_{t-1})$ and rearrange the terms; $(b)$ adds and subtracts $\frac{1-\kappa}{B} \sum_{\xi \in \mathcal{B}_t} (\nabla f(\mathbf{x}_t; \xi) - \nabla f(\mathbf{x}_{t-1}; \xi))$; $(c)$ directly expands the square and use the fact that $\mathbb{E}_t[\mathbf{w}_t] = 0$ and $\mathbf{w}_t$ is independent of other terms; and in $(d)$, we notice that $\mathbb{E}_t[D_1] = 0, \mathbb{E}_t[D_2] = 0$, so the first two inner products are zero, and we apply (19) to the other four inner products, with $\alpha = \kappa|1+\gamma|, 1, |1+\gamma|$, respectively. Next, we bound each term separately. For $\mathbb{E}_t[\|D_1\|^2]$, we have:

$$\mathbb{E}_t[\|D_1\|^2] = \mathbb{E}_t \left\| \nabla F(\mathbf{x}_t) - \nabla F(\mathbf{x}_{t-1}) - \frac{1}{B} \sum_{\xi \in \mathcal{B}_t} (\nabla f(\mathbf{x}_t; \xi) - \nabla f(\mathbf{x}_{t-1}; \xi)) \right\|^2$$

$$\overset{(a)}{\leq} \mathbb{E}_t \left\| \frac{1}{B} \sum_{\xi \in \mathcal{B}_t} (\nabla f(\mathbf{x}_t; \xi) - \nabla f(\mathbf{x}_{t-1}; \xi)) \right\|^2 \tag{23}$$

$$\overset{(b)}{\leq} L^2 \eta^2 \|\tilde{\mathbf{g}}_{t-1}\|^2$$

$$\overset{(c)}{\leq} 2L^2 \eta^2 (\|\Delta_{t-1}\|^2 + \|\nabla F(\mathbf{x}_{t-1})\|^2),$$

where $(a)$ uses the fact that $\mathbb{E}\|X - \mathbb{E}[X]\|^2 \leq \mathbb{E}\|X\|^2$, with $\nabla F(\mathbf{x}_t) - \nabla F(\mathbf{x}_{t-1}) = \mathbb{E}_t[\frac{1}{B}\sum_{\xi \in \mathcal{B}_t}(\nabla f(\mathbf{x}_t; \xi) - \nabla f(\mathbf{x}_{t-1}; \xi))]$; $(b)$ applies A1 and $(c)$ adds and subtracts $\nabla F(\mathbf{x}_{t-1})$, and applies (20). For $\mathbb{E}_t[\|D_2\|^2]$, we have:

$$\mathbb{E}_t[\|D_2\|^2] = \mathbb{E}_t \left\| \nabla F(\mathbf{x}_t) - \frac{1}{B}\sum_{\xi \in \mathcal{B}_t} \nabla f(\mathbf{x}_t, \xi) \right\|^2 \overset{A2}{\leq} \frac{\sigma_{\mathrm{SGD}}^2}{B}. \tag{24}$$

For $\mathbb{E}_t[\|D_3\|^2]$, we have:

$$\mathbb{E}_t[\|D_3\|^2] = \mathbb{E}_t \left\| \frac{1}{B}\sum_{\xi \in \mathcal{B}_t} \left( \frac{1}{\gamma}\nabla f(\mathbf{x}_t + \gamma \mathbf{d}_{t-1}; \xi) + \nabla f(\mathbf{x}_{t-1}; \xi) - \frac{1+\gamma}{\gamma}\nabla f(\mathbf{x}_t; \xi) \right) \right\|^2$$

$$\overset{(a)}{\leq} \frac{1}{B}\sum_{\xi \in \mathcal{B}_t} \left\| \frac{1}{\gamma}\nabla f(\mathbf{x}_t + \gamma \mathbf{d}_{t-1}; \xi) - \frac{1}{\gamma}\nabla f(\mathbf{x}_{t-1}; \xi) \right.$$

$$\left. + \frac{1+\gamma}{\gamma}\nabla f(\mathbf{x}_{t-1}; \xi) - \frac{1+\gamma}{\gamma}\nabla f(\mathbf{x}_t; \xi) \right\|^2$$

$$\overset{(20)}{\leq} \frac{2}{B}\sum_{\xi \in \mathcal{B}_t} \left\| \frac{1}{\gamma}\nabla f(\mathbf{x}_t + \gamma \mathbf{d}_{t-1}; \xi) - \frac{1}{\gamma}\nabla f(\mathbf{x}_{t-1}; \xi) \right\|^2$$

$$+ \frac{2}{B}\sum_{\xi \in \mathcal{B}_t} \left\| \frac{1+\gamma}{\gamma}\nabla f(\mathbf{x}_{t-1}; \xi) - \frac{1+\gamma}{\gamma}\nabla f(\mathbf{x}_t; \xi) \right\|^2$$

$$\overset{A1}{\leq} \frac{2L^2}{\gamma^2}\|\mathbf{x}_t + \gamma \mathbf{d}_{t-1} - \mathbf{x}_{t-1}\|^2 + \frac{2L^2(1+\gamma)^2}{\gamma^2}\|\mathbf{x}_{t-1} - \mathbf{x}_t\|^2$$

$$\overset{(b)}{=} \frac{4L^2(1+\gamma)^2\eta^2}{\gamma^2}\|\tilde{\mathbf{g}}_{t-1}\|^2$$

$$\overset{(c)}{\leq} \frac{8L^2(1+\gamma)^2\eta^2}{\gamma^2}(\|\Delta_{t-1}\|^2 + \|\nabla F(\mathbf{x}_{t-1})\|^2), \tag{25}$$

where $(a)$ applies Jensens' inequality to $\|\cdot\|^2$, and we add and subtract $\frac{1}{\gamma}\nabla f(\mathbf{x}_{t-1}; \xi)$; $(b)$ apples (20); $(b)$ uses the fact that $\mathbf{d}_{t-1} = \mathbf{x}_t - \mathbf{x}_{t-1} = -\eta\tilde{\mathbf{g}}_{t-1}$; and $(c)$ adds and subtracts $\nabla F(\mathbf{x}_{t-1})$, and applies (20). Plug in (23) – (25) to (22), we have:

$$\mathbb{E}_t\|\Delta_t\|^2 \leq (1-\kappa)^2(2 + |1+\gamma|)\mathbb{E}_t\|D_1\|^2 + (1-\kappa)^2(1 + \kappa|1+\gamma|)\|\Delta_{t-1}\|^2 + \kappa^2 d\sigma_{\mathrm{DP}}^2$$

$$+ \kappa^2(2 + |1+\gamma|)\mathbb{E}_t[\|D_2\|^2] + (1-\kappa)^2\left(1 + \frac{2}{|1+\gamma|} + \frac{1}{\kappa|1+\gamma|}\right)\mathbb{E}_t\|D_3\|^2$$

$$\leq (1-\kappa)^2\left(1 + 4\eta^2 L^2 + |1+\gamma|\left(\kappa + 2\eta^2 L^2 M_\gamma\right)\right)\|\Delta_{t-1}\|^2$$

$$+ 2\eta^2 L^2(1-\kappa)^2\left(2 + |1+\gamma| M_\gamma\right)\|\nabla F(\mathbf{x}_{t-1})\|^2$$

$$+ \kappa^2\left((2 + |1+\gamma|)\frac{\sigma_{\mathrm{SGD}}^2}{B} + d\sigma_{\mathrm{DP}}^2\right), \tag{26}$$

where we define $M_\gamma := \left(1 + \frac{4(2+1/\kappa+|1+\gamma|)}{\gamma^2}\right)$. This completes the proof of Lemma 1.

## B.2 PROOF OF THEOREM 2

Now, we are ready to prove Theorem 2. By choosing $C \geq G(1 + \frac{2(1-\kappa)}{\kappa\gamma})$, the clipping is inactive. Recall that from the update rule, we have

$$\mathbb{E}_t[\tilde{\mathbf{g}}_t] = (1-\kappa)\tilde{\mathbf{g}}_{t-1} + \kappa\nabla F(\mathbf{x}_t) + (1-\kappa)(\nabla F(\mathbf{x}_t) - \nabla F(\mathbf{x}_{t-1}))$$

$$= \nabla F(\mathbf{x}_t) - (1-\kappa)\Delta_{t-1}. \tag{27}$$

Then, by A1 we have $F(\cdot)$ is also $L$-smooth, so it satisfies

$$F(\mathbf{y}) \leq F(\mathbf{x}) + \langle \nabla F(\mathbf{x}), \mathbf{y} - \mathbf{x} \rangle, \frac{L}{2}\|\mathbf{y} - \mathbf{x}\|^2.$$

Substitute $\mathbf{y} = \mathbf{x}_{t+1}, \mathbf{x} = \mathbf{x}_t$ to the above relation and take expectation over the randomness in iteration $t$, we have:

$$\mathbb{E}_t[F(\mathbf{x}_{t+1})] - F(\mathbf{x}_t) \leq -\eta \langle \nabla F(\mathbf{x}_t), \mathbb{E}_t[\tilde{\mathbf{g}}_t] \rangle + \frac{\eta^2 L}{2} \mathbb{E}_t \|\tilde{\mathbf{g}}_t\|^2$$

$$\stackrel{(a)}{=} -\eta \|\nabla F(\mathbf{x}_t)\|^2 + \eta(1-\kappa) \langle \nabla F(\mathbf{x}_t), \Delta_{t-1} \rangle + \frac{\eta^2 L}{2} \mathbb{E}_t \|\tilde{\mathbf{g}}_t\|^2$$

$$\stackrel{(19)}{\leq} -\eta \|\nabla F(\mathbf{x}_t)\|^2 + \frac{\eta(1-\kappa)}{2} \|\nabla F(\mathbf{x}_t)\|^2 + \frac{\eta(1-\kappa)}{2} \|\Delta_{t-1}\|^2 + \frac{\eta^2 L}{2} \mathbb{E}_t \|\tilde{\mathbf{g}}_t\|^2$$

$$\stackrel{(b)}{=} -\eta \|\nabla F(\mathbf{x}_t)\|^2 + \frac{\eta(1-\kappa)}{2} \|\nabla F(\mathbf{x}_t)\|^2 + \frac{\eta(1-\kappa)}{2} \|\Delta_{t-1}\|^2$$

$$+ \frac{\eta^2 L}{2} \mathbb{E}_t \|\tilde{\mathbf{g}}_t - \nabla F(\mathbf{x}_t) + \nabla F(\mathbf{x}_t)\|^2$$

$$\stackrel{(c)}{\leq} -\frac{\eta(1+\kappa-2\eta L)}{2} \|\nabla F(\mathbf{x}_t)\|^2 + \frac{\eta(1-\kappa)}{2} \|\Delta_{t-1}\|^2 + \eta^2 L \, \mathbb{E}_t \|\Delta_t\|^2, \qquad (28)$$

where $(a)$ substitute (27); $(b)$ we add and subtract $\nabla F(\mathbf{x}_t)$ to the last term and $(c)$ applies (20) to the last term.

Define $\mathcal{L}_t := F(\mathbf{x}_t) + \beta \|\Delta_{t-1}\|^2$, we have:
$$\mathbb{E}_t[\mathcal{L}_{t+1}] - \mathcal{L}_t$$

$$\leq -\frac{\eta(1+\kappa-2\eta L)}{2} \|\nabla F(\mathbf{x}_t)\|^2 - (\beta - \frac{\eta(1-\kappa)}{2}) \|\Delta_{t-1}\|^2 + (\beta + \eta^2 L) \, \mathbb{E}_t \|\Delta_t\|^2$$

$$\stackrel{(a)}{\leq} -\frac{\eta(1+\kappa-2\eta L)}{2} \|\nabla F(\mathbf{x}_t)\|^2 - (\beta - \frac{\eta(1-\kappa)}{2}) \|\Delta_{t-1}\|^2$$

$$+ (\beta + \eta^2 L)(1-\kappa)^2(1 + 4L^2\eta^2 + |1+\gamma| \, (\kappa + 2\eta^2 L^2 M_\gamma)) \|\Delta_{t-1}\|^2$$

$$+ (\beta + \eta^2 L)\kappa^2 \left( \frac{(2+|1+\gamma|)\sigma_{\text{SGD}}^2}{B} + d\sigma_{\text{DP}}^2 \right)$$

$$+ 2(\beta + \eta^2 L)(1-\kappa)^2 L^2\eta^2 \, (2 + |1+\gamma| \, M_\gamma) \|\nabla F(\mathbf{x}_{t-1})\|^2$$

$$= -\frac{\eta(1+\kappa-2\eta L)}{2} \|\nabla F(\mathbf{x}_t)\|^2 + 2(\beta + \eta^2 L)(1-\kappa)^2 L^2\eta^2 \, (2 + |1+\gamma| \, M_\gamma) \|\nabla F(\mathbf{x}_{t-1})\|^2$$

$$- \left( \beta - \frac{\eta(1-\kappa)}{2} - (\beta + \eta^2 L)(1-\kappa)^2(1 + 4L^2\eta^2 + |1+\gamma| \, (\kappa + 2\eta^2 L^2 M_\gamma)) \right) \|\Delta_{t-1}\|^2$$

$$+ (\beta + \eta^2 L)\kappa^2 \left( \frac{(2+|1+\gamma|)\sigma_{\text{SGD}}^2}{B} + d\sigma_{\text{DP}}^2 \right), \qquad (29)$$

where $(a)$ applies Lemma 1 and $(b)$ rearrange the terms. By choosing

$$\eta < \frac{1}{2L(1 + 2(1-\kappa)^2 \beta L(2 + |1+\gamma| \, M_\gamma))}, \quad \kappa > 1 - \frac{1}{\sqrt{1 + 4\eta^2 L^2 + |1+\gamma| \, (\kappa + 2\eta^2 L^2 M_\gamma)}},$$

$$\beta \geq \frac{\eta(1-\kappa)/2 + \eta^2 L(1-\kappa)^2(1 + 4\eta^2 L^2 + |1+\gamma| \, (\kappa + 2\eta^2 L^2 M_\gamma))}{1 - (1-\kappa)^2(1 + 4\eta^2 L^2 + |1+\gamma| \, (\kappa + 2\eta^2 L^2 M_\gamma))},$$

we have:
$$\frac{\eta(1+\kappa-2\eta L)}{2} - 2(\beta + \eta^2 L)(1-\kappa)^2 L^2\eta^2 \, (2 + |1+\gamma| \, M_\gamma) > 0,$$

$$1 - (1-\kappa)^2(1 + 4\eta^2 L^2 + |1+\gamma| \, (\kappa + 2\eta^2 L^2 M_\gamma)) > 0, \qquad (30)$$

$$\beta - \frac{\eta(1-\kappa)}{2} - (\beta + \eta^2 L)(1-\kappa)^2(1 + 4L^2\eta^2 + |1+\gamma| \, (\kappa + 2\eta^2 L^2 M_\gamma)) \geq 0.$$

Average from $t = 0$ to $T - 1$ and rearrange the terms, we have:

$$\frac{1}{T} \sum_{t=0}^{T} \mathbb{E} \|\nabla F(\mathbf{x}_t)\|^2 \leq \frac{2(\mathcal{L}_0 - \mathbb{E}[\mathcal{L}_{T+1}])}{M_1 \eta T} + \frac{2(\beta + \eta^2 L)\kappa^2}{M_1 \eta} \left( \frac{(2+|1+\gamma|)\sigma_{\text{SGD}}^2}{B} + d\sigma_{\text{DP}}^2 \right)$$

$$\leq \frac{2(F(\mathbf{x}_0) + \beta \|\nabla F(\mathbf{x}_0)\|^2 - F^\star)}{M_1 \eta T} + \frac{2(\beta + \eta^2 L)\kappa^2}{M_1 \eta} \left( \frac{(2+|1+\gamma|)\sigma_{\text{SGD}}^2}{B} + d\sigma_{\text{DP}}^2 \right), \quad (31)$$

where we define $M_1 := (1 + \kappa - 2\eta L) - 4(\beta + \eta^2 L)(1 - \kappa)^2 L^2 \eta (2 + |1 + \gamma| M_\gamma)$, and in the last inequality we notice that $\mathcal{L}_{T+1} = F(\mathbf{x}_{T+1}) + \beta \|\Delta_T\|^2 \geq F^\star$, and $\mathcal{L}_0 = F(\mathbf{x}_0) + \beta \|\nabla F(\mathbf{x}_0)\|^2$, as we initialize $\tilde{\mathbf{g}}_0 = 0$. This completes the proof of Theorem 2.

**On the choice of** $\gamma = -1$**.** From the above proof, we see that in Appendix B.1, (22) $(c)$, we directly apply (19) to upper bound the cross-product terms by positive terms for the worst case, which results in the optimal choice of $\gamma = -1$. However, the inner products may be smaller than zero in some cases, making $\gamma = -1$ sub-optimal in practice.

## C  ADDITIONAL NUMERICAL RESULTS

### C.1  EXPERIMENT DETAILS

**Coding:** The code for the experiments will be provided online. We use PyTorch as the code base and the FastDP package (Bu et al., 2023) to privatize the optimizers. We use the Renyi differential privacy (RDP) accountant in the Opacus and FastDP packages to numerically calculate the required injected DP noise to the gradient for fixed $(\epsilon, \delta)$-DP budget. A detailed derivation of the RDP accountant can be found in Wang et al. (2019). The Algorithm 3 is implemented as a PyTorch optimizer, which can be easily combined with any training scripts based on PyTorch. The modification is minimum:

```
1  from KFOptimizer import wrap_optimizer
2  # define base optimizer
3  optimizer = wrap_optimizer(base_optimizer, kappa, gamma)
4  # ...
5  # in training loop:
6      def closure(): # warp up the loss and backward computation
7          loss = model(input)
8          loss.backward()
9          return loss
10     loss = optimizer.prestep(closure)
11     # ...
```

The link to the full code of the experiments can be found at `https://anonymous.4open.science/r/KalmanDP-BEDB`. Alternative implementation in Opacus can be found at `https://github.com/pytorch/opacus/tree/main/research/disk_optimizer`.

**Hardware:** All the experiments except the ImageNet-1k dataset are running with one Nvidia A40 (48GB memory) or one Nvidia V100 (32GB memory). The experiment on the ImageNet-1k dataset is running on one Nvidia H100 (80GB memory) GPU. The training time varies for different tasks depending on the data size and model size.

**Training method:** We use gradient accumulation to deal with the large batch size and use learning rate warm-up for $1/20$ of the training steps when training from randomly initialized weights. We also use the Cosine Annealing learning rate scheduler (Loshchilov & Hutter, 2022), which gradually decreases the learning rate.

### C.2  CHOICE OF HYPER-PARAMETERS

The main hyper-parameters in the algorithms are: epoch $E$, batch size $B$, step size $\eta$, clipping threshold $C$, Kalman filter parameters $\kappa$, and $\gamma$. In all experiments, we fix the clipping method as automatic clipping used in Bu et al. (2024), i.e., $\text{clip}(\nabla f(\mathbf{x}; \xi), C) = \nabla f(\mathbf{x}; \xi) \cdot \frac{C}{\|\nabla f(\mathbf{x}; \xi)\|}$, and set $C = 1$ for all experiments. We fix $\delta = 1/N^{1.1}$ for reasonable privacy notions. This choice matches or is tighter than the SOTA results using $\delta = 1/2N$ or $1/N^{1.1}$ (Li et al., 2021; Bu et al., 2024; Yu et al., 2021). We list the $\delta$'s used in the experiments in Tab. 7.

For each set of experiments, we conduct a grid search on the hyper-parameters $E, B, \eta$ and choose the optimal ones for the DP optimizer without DiSK based on the test set. The search grids of each hyper-parameter are listed in Table 2;

Then, we fix $E, B.\eta$ and conduct the ablation study on $\kappa, \gamma$ as shown in Figure 8.

Table 2: Search grid of the CV pre-training experiments, the optimal hyper-parameters are in **bold**.

| | Search gird | | |
|---|---|---|---|
| | MNIST | CIFAR | ImageNet |
| $E$ | $\{1, \mathbf{2}, 3\} \times 20$ | $\{1, \mathbf{2}, 3, 4\} \times 40$ | $\{3, \mathbf{4}\} \times 40$ |
| $B$ | $\{2, \mathbf{5}\} \times 10^3$ | $\{0.5, 1, 2, \mathbf{5}\} \times 10^3$ | $\{\mathbf{5}, 10\} \times 10^3$ |
| $\eta$ | $\{3, \mathbf{2.5}, 1, 0.3, 0.1\} \times 10^{-1}$ | $\{1, 3, \mathbf{5}, 7, 10\} \times 10^{-3}$ | $\{10, 3, 1, \mathbf{0.3}, 0.1\} \times 10^{-3}$ |
| $\kappa$ | $\{\mathbf{0.7}\}$ | $\{9.9, 9, 8, \mathbf{7}, 6, 5\} \times 10^{-1}$ | $\{\mathbf{0.7}\}$ |
| $\gamma$ | $\{\mathbf{0.5}\}$ | $\{0.2, 0.3, \mathbf{0.5}, 1, 2, 3, \frac{1-\kappa}{\kappa}\}$ | $\{\mathbf{0.5}\}$ |

## C.3 ADDITIONAL EXPERIMENTS ON CV TASKS

**Training different models from scratch:** We additionally train the WRN-16-4 and the ViT-small models on the CIFAR-10 with randomly initialized weights, and the test accuracies during the training are shown in Figure 4 for $\epsilon = 4$. From Figure 4, we can see that DiSK consistently outperforms the base optimizer.

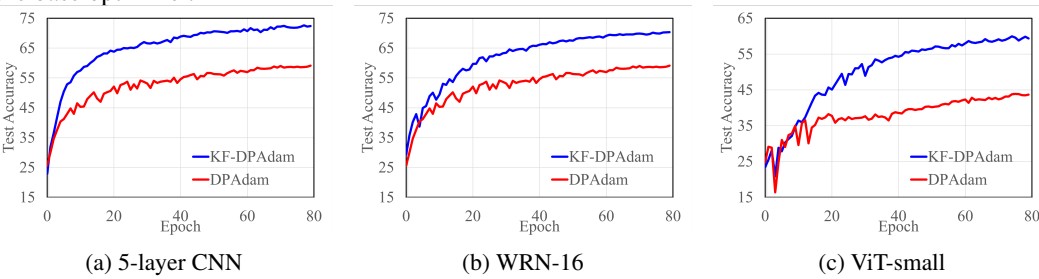

(a) 5-layer CNN                (b) WRN-16                (c) ViT-small

Figure 4: Test accuracy of pre-training 5-layer CNN, WRN-16, and ViT-small on CIFAR-10 dataset with and without DiSK for fixed privacy budget $\epsilon = 4$.

**Fine-tuning on CIFAR-100:** Besides training from scratch, we also compare the performance of fine-tuning a pre-trained ViT-small model on the CIFAR-100 dataset. The results for different $\epsilon$ are shown in Figure 5. For fine-tuning on the complex CIFAR-100 dataset, DiSK still improves the performance compared with DPAdam, and has less performance drop under large DP noise.

**Comparison with existing methods and SOTA:** We conduct comparisons with existing approaches for improving DP training performance. In Figure 6a, we train a linear regression model with synthetic data and compared the performance of Noisy GD, Noisy GD with DOPPLER (NoisyLP), and with DiSK (NoisyKF). We inject Gaussian noise with different variances into the gradient and compare the final performance. We observe that DiSK has the lowest regression loss under all noise levels, indicating that the Kalman filter performs better in noise reduction than the Low-pass filter. In Figure 6b, we compare the test accuracy of different methods, including DOPPLER (Zhang et al., 2024a) and DP-FTRL (Kairouz et al., 2021; Choquette-Choo et al., 2024) on the CIFAR-10 dataset training the WRN from scratch. We observe that DiSK significantly outperforms the SOTA algorithms on all privacy budgets. As discussed Section 3, DOPPLER refers to the case that DiSK only

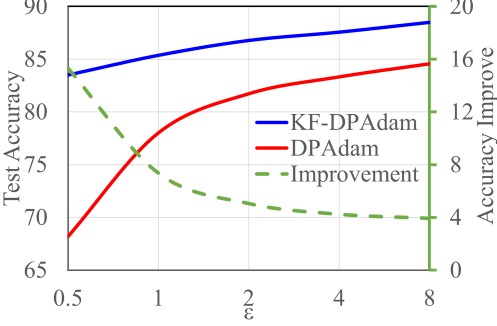

Figure 5: Fine-tuning ViT-small on CIFAR-100 with different $\epsilon$. The green line indicates the improvement of DiSK.

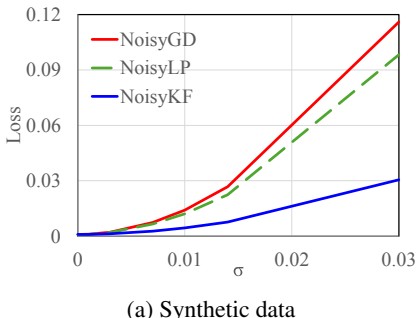 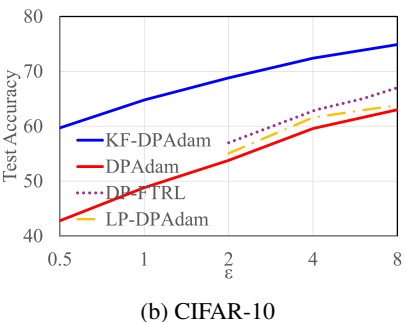

(a) Synthetic data                          (b) CIFAR-10

Figure 6: Comparison with existing approaches. a) Kalman filter and low-pass filter; b) Kalman filter, DP-FTRL, and low-pass filter.

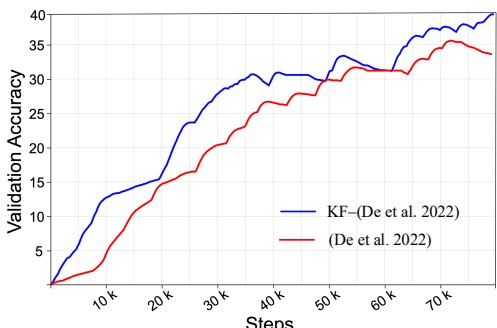

Figure 7: Comparison with the SOTA approach in De et al. (2022) on the ImageNet-1k dataset.

estimates gradient at one point $\mathbf{x}_t$ instead of performing a weighted average of gradient estimated at two points $\mathbf{x}_t, \mathbf{x}_t + \gamma \mathbf{d}_{t-1}$ before clipping. The results in Figure 6a and Figure 6b indicate that the *weighted average before clipping* may be more important (i.e., $\gamma \neq 0$ is more important).

Comparison with SOTA on ImageNet: Additionally, we conduct experiments to compare the performance of DiSK with SOTA result training from scratch on the ImageNet-1k dataset in De et al. (2022). The result is shown in Figure 7. We note that the reported test accuracy in the De et al. (2022) is 32.4% for privacy level $\epsilon = 8$. However, we noticed that their accounting procedure has been improved in their GitHub repository and hence we re-ran their experiment using the new accountant to obtain our baseline. We kept all their options on, including group normalization, larger batch size, weight standardization, augmentation multiplicity, and model exponential moving average. In addition, we also follow their *normalized clipping* strategy, i.e.,

$$\text{clip}\left(\nabla f, C\right) = \frac{\nabla f}{C} \min\left\{\frac{C}{\|\nabla f\|}, 1\right\}.$$

Their updated code results in 36.35% validation accuracy and the test accuracy of 33.56% (as the SOTA result). In comparison, adding DiSK on top of their method results in a validation accuracy of 40% (3.7% improvement) and a test accuracy of 36.89% (3.3% improvement). This sets a new SOTA result for differentially private training on the ImageNet-1k dataset.

**Ablation study:** We conduct ablation studies on the choice of the hyper-parameters of DiSK, specifically, how $\kappa, \gamma$ impact the algorithm performance. In Figure 8, we plot the accuracy on different combinations of $(\kappa, \gamma)$, and $(\kappa, \epsilon)$. We observe a clear trend of performance change for different combinations of the parameters, and there is an optimal choice of $\kappa, \gamma$ for different $\epsilon$'s.

## C.4 Additional experiments on NLP tasks

In this section, we provide additional results for NLP tasks.

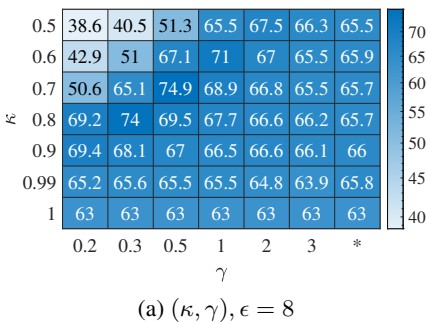

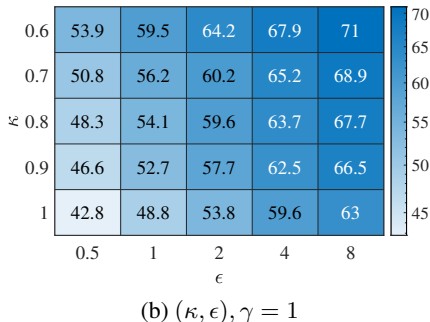

(a) $(\kappa, \gamma), \epsilon = 8$

(b) $(\kappa, \epsilon), \gamma = 1$

Figure 8: Test accuracy for different combinations of the hyper-parameters when training CNN on the CIFAR-10 dataset.

**Parameter-efficient fine-tuning on GLUE.** We fine-tune a RoBERTa-base and a RoBERTa-large model from the Huggingface checkpoints[1] on the GLUE dataset. We follow the same training scripts in Bu et al. (2024) on the hyper-parameter choices of $\eta, B, E$ on the tasks and use rank $r = 16$ for LoRA. We choose $\kappa = 0.7, \gamma = 0.5$ for DiSK. The results are listed in Table 3. With privacy budget $\epsilon = 1, 6.7$, DPLoRA with DiSK significantly outperforms SOTA results with vanilla DPLoRA on all tasks.

Table 3: Test accuracy of fine-tuning result on the GLUE dataset.

| Algorithm | $\epsilon = 1$ | | | | $\epsilon = 6.7$ | | | |
|---|---|---|---|---|---|---|---|---|
| | MNLI | QNLI | SST2 | QQP | MNLI | QNLI | SST2 | QQP |
| | RoBERTa-base | | | | | | | |
| AdamW ($\epsilon = \infty$) | 87.6 | 92.8 | 94.8 | 91.9 | 87.6 | 92.8 | 94.8 | 91.9 |
| Lora ($\epsilon = \infty$) | 87.5 | 93.3 | 95.1 | 90.8 | 87.5 | 93.3 | 95.1 | 90.8 |
| DPLora | 81.1 | 85.5 | 90.9 | 83.9 | 83.5 | 87.4 | 91.5 | 85.7 |
| **KF-DPLora** | 84.7 | 90.3 | 92.9 | 87.8 | 85.9 | 90.5 | 93.1 | 89.0 |
| | RoBERTa-large | | | | | | | |
| AdamW ($\epsilon = \infty$) | 90.3 | 94.7 | 96.4 | 92.2 | 90.3 | 94.7 | 96.4 | 92.2 |
| Lora ($\epsilon = \infty$) | 90.6 | 94.9 | 96.2 | 91.6 | 90.6 | 94.9 | 96.2 | 91.6 |
| DPLora | 85.6 | 89.5 | 90.9 | 85.1 | 87.8 | 90.8 | 94.3 | 87.4 |
| **KF-DPLora** | 87.9 | 92.5 | 95.2 | 88.2 | 89.4 | 92.6 | 95.4 | 89.6 |

**Fine-tuning GPT-2 on text generation tasks.** We fine-tune a GPT-2-small model with $137M$ parameters from the Huggingface checkpoints[2] on two text generation datasets, E2E and DART. We follow the same training scripts in Li et al. (2021) on the hyper-parameter choices of $\eta, B, E$ on the tasks and choose $\kappa = 0.7, \gamma = 0.5$ for DiSK. The results on different metrics for the E2E dataset are given in Table 4, and the results for the DART dataset are in Table 5. With privacy budget $\epsilon = 3, 8$, DPAdamW with DiSK significantly outperforms SOTA results with vanilla DPAdamW on all metrics.

Table 4: Performance of fine-tuning gpt-2 on the E2E dataset. (All metrics are higher the better)

| Algorithm | BLEU (%) | ROUGE-L (%) | METEOR | NIST | CIDEr |
|---|---|---|---|---|---|
| AdamW ($\epsilon = \infty$) | 69.46 | 71.36 | 0.461 | 8.780 | 2.422 |
| DPAdamW ($\epsilon = 3$) | 61.52 | 65.87 | 0.417 | 7.071 | 2.167 |
| **KF-DPAdamW ($\epsilon = 3$)** | 68.35 | 70.23 | 0.456 | 8.636 | 2.399 |
| DPAdamW ($\epsilon = 8$) | 64.99 | 67.34 | 0.425 | 8.387 | 2.192 |
| **KF-DPAdamW ($\epsilon = 8$)** | 68.73 | 70.58 | 0.460 | 8.697 | 2.463 |

---

[1]https://huggingface.co/FacebookAI/roberta-base,https://huggingface.co/FacebookAI/roberta-large

[2]https://huggingface.co/openai-community/gpt2

Table 5: Performance of fine-tuning gpt-2 on the DART dataset. Val. Perp. stands for validation perplexity. (All metrics except Val. Perp. are higher the better)

| Algorithm | Val. Perp. ↓ | BLEU (%) | ROUGE-L (%) | METEOR | NIST | CIDEr |
|---|---|---|---|---|---|---|
| AdamW ($\epsilon = \infty$) | 0.921 | 44.56 | 58.66 | 0.379 | 8.733 | 2.773 |
| DPAdamW ($\epsilon = 3$) | 1.427 | 33.96 | 52.38 | 0.310 | 6.090 | 1.864 |
| **KF-DPAdamW** ($\epsilon = 3$) | 1.149 | 41.01 | 57.53 | 0.359 | 7.949 | 2.553 |
| DPAdamW ($\epsilon = 8$) | 1.362 | 35.30 | 54.58 | 0.320 | 6.365 | 1.995 |
| **KF-DPAdamW** ($\epsilon = 8$) | 1.102 | 42.12 | 58.11 | 0.364 | 8.111 | 2.628 |

## C.5 IMPROVEMENT OVER SOTA

In the following Table 6, we provide a list of the experiment settings where our algorithm outperforms SOTA results. The paper that provides the SOTA results is cited for each setting, or the same as the line above. We observe that DiSK provides new SOTA for CV datasets, including CIFAR-10, CIFAR-100, and ImageNet-1k, and NLP datasets, including GLUE (MNLI, QNLI, QQP, SST-2), DART, and E2E.

Table 6: Tasks DiSK improves SOTA, PT=pre-training, FT=fine-tuning.

| Dataset | TASK | Model | $\epsilon$ | Ours (%) | SOTA (%) |
|---|---|---|---|---|---|
| CIFAR-10 | PT | CNN | 0.5 | 59.7 | N/A |
| CIFAR-10 | PT | CNN | 2 | 68.8 | 67.2 (Tramer & Boneh, 2020) |
| CIFAR-100 | PT | WRN | 0.5 | 14.7 | N/A |
| CIFAR-100 | PT | WRN | 1 | 22.7 | 14.1 (Bao et al., 2024) |
| CIFAR-100 | PT | WRN | 2 | 30.0 | 21.5 |
| CIFAR-100 | PT | WRN | 4 | 37.1 | 33.3 |
| CIFAR-100 | PT | WRN | 8 | 42.0 | 40.6 |
| CIFAR-100 | FT | ViT | 0.5 | 83.49 | 78.3 (Mehta et al., 2023) |
| CIFAR-100 | FT | ViT | 1 | 85.36 | 81.8 (Bao et al., 2024) |
| CIFAR-100 | FT | ViT | 2 | 86.77 | 83.5 |
| CIFAR-100 | FT | ViT | 4 | 87.56 | 84.5 |
| CIFAR-100 | FT | ViT | 8 | 88.49 | 84.6 |
| ImageNet-1k | PT | ResNet-50 | 8 | 36.89 | 33.56 (De et al., 2022) |
| MNLI | FT | RoBERTa-base | 1 | 84.7 | 83.2 ($\epsilon = 3$) (Bu et al., 2023) |
| QNLI | FT | RoBERTa-base | 1 | 90.3 | 87.4 ($\epsilon = 3$) |
| QQP | FT | RoBERTa-base | 1 | 87.8 | 85.8 ($\epsilon = 3$) |
| SST-2 | FT | RoBERTa-base | 1 | 92.9 | 92.3 ($\epsilon = 3$) |
| MNLI | FT | RoBERTa-base | 6.7 | 85.9 | 83.8 ($\epsilon = 8$) (Bu et al., 2023) |
| QNLI | FT | RoBERTa-base | 6.7 | 90.5 | 87.9 ($\epsilon = 8$) |
| QQP | FT | RoBERTa-base | 6.7 | 89.0 | 86.6 ($\epsilon = 8$) |
| SST-2 | FT | RoBERTa-base | 6.7 | 93.1 | 93.0 ($\epsilon = 8$) (Li et al., 2021) |
| MNLI | FT | RoBERTa-large | 1 | 87.9 | 86.8 (Yu et al., 2021) |
| QNLI | FT | RoBERTa-large | 1 | 92.5 | 88.0 |
| QQP | FT | RoBERTa-large | 1 | 88.2 | 85.2 |
| SST-2 | FT | RoBERTa-large | 1 | 95.2 | 93.1 |
| MNLI | FT | RoBERTa-large | 6.7 | 89.4 | 89.0 (Yu et al., 2021) |
| QNLI | FT | RoBERTa-large | 6.7 | 92.6 | 92.5 |
| QQP | FT | RoBERTa-large | 6.7 | 89.6 | 88.4 |
| SST-2 | FT | RoBERTa-large | 6.7 | 95.4 | 95.3 |
| E2E (BLEU) | FT | GPT-2 | 3 | 68.35 | 61.52 (Li et al., 2021) |
| E2E (ROUGE-L) | FT | GPT-2 | 3 | 70.23 | 65.87 (Bu et al., 2024) |
| E2E (BLEU) | FT | GPT-2 | 8 | 68.73 | 63.60 (Bu et al., 2024) |
| E2E (ROUGE-L) | FT | GPT-2 | 8 | 70.58 | 67.53 (Li et al., 2021) |
| DART (BLEU) | FT | GPT-2 | 3 | 41.01 | 32.33 (Li et al., 2021) |
| DART (ROUGE-L) | FT | GPT-2 | 3 | 57.53 | 52.06 |
| DART (BLEU) | FT | GPT-2 | 8 | 42.12 | 35.06 |
| DART (ROUGE-L) | FT | GPT-2 | 8 | 58.11 | 54.57 |

Table 7: The privacy parameter $\delta$'s used in our experiments and in SOTA results.

| Dataset | Our $\delta$ | SOTA $\delta$ |
|---|---|---|
| MNIST | $5.5 \times 10^{-6}$ | $10^{-5}$ |
| CIFAR-10/100 | $6.8 \times 10^{-6}$ | $10^{-5}$ |
| Imagenet-1k | $1.9 \times 10^{-7}$ | $8 \times 10^{-7}$ |
| MNLI | $6.3 \times 10^{-7}$ | $1.1 \times 10^{-6}$ |
| QNLI | $4.8 \times 10^{-7}$ | $9 \times 10^{-7}$ |
| SST-2 | $4.9 \times 10^{-6}$ | $7.4 \times 10^{-6}$ |
| QQP | $7.6 \times 10^{-7}$ | $1.4 \times 10^{-6}$ |
| E2E | $1.2 \times 10^{-5}$ | $1.2 \times 10^{-5}$ |
| DART | $6.1 \times 10^{-6}$ | $6.1 \times 10^{-6}$ |

