# OpenReview forum: "DiSK: Differentially Private Optimizer with Simplified Kalman Filter for Noise Reduction"
_ICLR.cc/2025/Conference — ICLR 2025 Poster_

### Official Review · Reviewer_PnV2 · 2024-11-01

**Soundness:** 3
**Presentation:** 4
**Contribution:** 3
**Rating:** 8
**Confidence:** 4

**Summary:**

This paper proposes DiSK. DiSK is inspired by an application of a Kalman filter to the DP-SGD process. While the Kalman filter includes computationally intensive steps and requires knowledge of some unknown matrices, the simplification steps result in DP-SGD with two seemingly small changes. The first is that the clipped gradient is computed from an average of the gradient at two carefully chosen nearby points. The second is a momentum applied to the clipped gradient before being passed into an optimizer.

With these changes, the authors prove an upper bound on convergence rate. The provable constant in the upper bound for DiSK is smaller than the provable constant in the upper bound for vanilla SGD. Although this does not mean that the paper proved that DiSK has faster convergence (a claim that should be fixed), it does suggest that DiSK would converge faster. The analysis suggests that the pre-clipping averaging step may be the more important of the differences between DiSK and DP-SGD.

The paper also performs extensive experiments to compare against prior work to establish SOTA. It is not clear that the experiments match the deltas of the prior SOTA. I wasn't able to match the reported SOTA numbers to the papers that I checked, and some SOTA numbers are missing.

**Strengths:**

The paper reads very nicely. The authors take the reader through a tour of their intuition when explaining the algorithm. The theoretical results are also discussed nicely. Most papers simply present a Big-O result and switch topics. This paper explained the significance of the theorem conclusion.

I like the fact that the resulting algorithm is simple, increasing the chance of it being widely adopted.

The theoretical analysis is nice (theorem 2 and corollary 1) and shows that the provable convergence upper bound can be improved by a constant factor that depends on the model and data. It is not the same as claiming that faster convergence has been proved (Line 419, but that claim can be reworded to be accurate)

The model technique achieves state of the art in many settings (there is a small asterisk because I am not fully convinced yet, see below)..

**Weaknesses:**

There is a lot of ambiguity in the paper. Critical details are not discussed to ensure comparisons are apples to apples (see below).

There are discrepancies in the experiments and comparisons to prior SOTA (see below)

**Questions:**

My initial rating is a weak reject that I am willing to raise with good responses to these questions.

# Questions about ambiguity
- The way neighbors are defined has been left ambiguous. Are two datasets neighbors if adding/removing a record turns one into the other? Are two datasets neighbors if modifying one record turns one into another? The choice can result in significant differences in sensitivity and applicability of subsampling amplification results.

- What form of subsampling was used for the creation of minibatches. Simple random sample with a fixed size? Poisson sampling?

- What privacy accountant was used?

- Algorithms 1,2,3 should use parentheses to more clearly resolve the ambiguity of whether the noise term belongs
in the summation or outside.


# Experimental questions
- Did you match neighbor definitions in the comparisons to prior SOTA?

- Have you tried any of the privacy auditing frameworks (e.g., relatively recent NeurIPS papers) to check for possible errors? For example, depending on when you add noise and when you average over a batch, an error could easily appear to be SOTA and this is not uncommon in DP-SGD.

- The algorithms is 2 changes away from vanilla DP-SGD. The paper has hyperparmaeter tuning studies that are named ablation studies, but it does not have true ablation studies. There are 4 possible comparisons that can shed light on what is responsible for most of the apparent performance improvements. (1) vanilla dp-SGD (k=1, gamma=0); (2) weighted average before clipping, but no momentum on the gradient after clipping; (3) only momentum in the gradient and no pre-clipping averaging (gamma=0), (4) DiSK.
    - Corollary 1 implies that gamma may be the more important parameter.

- The paper does not explain how hyperparameters were selected. Were they chosen based on validation data (good) or testing data (thus favoring any approach that has more hyperparameters).

- I tried to match up some of the SOTA numbers in the appendix as a spot check.
    - Specifically, I looked at De et al. because they have a few prior SOTA numbers in the same place. I couldn't match their Imagenet numbers (32.4% top-1 accuracy to the 36.5 written in Appendix C5).
   - I also noticed that some of the high CIFAR-10 training from scratch numbers in De et al. are completely missing from the Appendix.
   - It is not clear to me whether DiSK is appropriately compared to the prior SOTA. Do you match deltas? Do you match neighbor definitions and batch subsampling? Do you match privacy accountants?
   - Delta is a big difference maker, so you should put its numerical value(s) in every table.
   - Table 6 does not match imagenet for table 2.

# Other questions/comments
- What is that wacky green line in Figure 1?
- The paper uses C for too many different purposes. In some cases it is a clipping threshold, other times a constant in theoretical bounds, and other times a constant on properties of the data. This can get confusing even with subscripts.
- Line 121: where is the delta < 0.05 restriction coming from? If I am not mistaken, Definition 2 is all Balle & Wang and does not use the results from Dwork and Roth.
-Line 086-089: such comparisons are incomplete without specifying privacy parameters and adding citations.

---

> ### Author Response · Authors · 2024-11-22
> **Response to Reviewer PnV2 Part I**
>
> Thank you for recognizing the novelty and strength of the theory and experiments in our paper. We are glad that you found our algorithm effective and simple enough to be widely adopted. Below is our response to your questions:
>
> Before answering your concerns in detail, we would like to clarify that our paper does not propose a new privacy analysis or new implementation of the DP mechanism. We use the public code bases of existing SOTA papers (Jax-privacy in De et al., Opacus in  Li et al., and FastDP in Bu et al.). We do not modify the privacy-preserving procedures used in the code of the SOTA experiments, including sampling, privacy accountant, clipping, and noise injection. Therefore, our numerical results can be fairly compared with the SOTA results.
>
> **Summary:**
> > ...  Although this does not mean that the paper proved that DiSK has faster convergence (a claim that should be fixed)
>
> Thanks for the comment. We have revised the statement of the theoretical improvement as the upper bound of iteration complexity has been improved by a constant factor.
>
> **Questions:**
>
> ### Questions about ambiguity
> > * The way neighbors are defined has been left ambiguous. Are two datasets neighbors if adding/removing a record turns one into the other? Are two datasets neighbors if modifying one record turns one into another? The choice can result in significant differences in sensitivity and applicability of subsampling amplification results.
>
> The theoretical analysis in Thm. 1 (Abadi et al.) is for neighboring datasets refers to adding/removing a record. This definition is also widely used in the theoretical analysis of DP in other papers (e.g., Wang et al. 2019 and Mironov et al. 2019).
>
> > * What form of subsampling was used for the creation of minibatches. Simple random sample with a fixed size? Poisson sampling?
>
> There exists theoretical analysis for both sampling without replacement and Poisson sampling (see Wang et al. 2019 and Mironov et al. 2019). In the experiment, we use sampling without replacement for all tasks. The privacy accountant (RDP) in the experiments provides the correct privacy budget for the used sampling procedure. We clarified this in our revised manuscript in the experiment setting.
>
> > * What privacy accountant was used?
>
> We use RDP, which is consistent with experiments in other papers that serve as baselines for our comparison (Bu et al., Li et al., De et al.)
>
> > * Algorithms 1,2,3 should use parentheses to more clearly resolve the ambiguity of whether the noise term belongs in the summation or outside.
>
> Thanks for the suggestion. $w_t$ is outside of the summation, and that is why there are no parentheses in Algs. 1,2,3.
>
> ### Experimental questions
> > * Did you match neighbor definitions in the comparisons to prior SOTA?
>
> Yes, the definitions of neighboring datasets are the same.
>
> > * Have you tried any of the privacy auditing frameworks  (e.g., relatively recent NeurIPS papers) to check for possible errors? For example, depending on when you add noise and when you average over a batch, an error could easily appear to be SOTA and this is not uncommon in DP-SGD.
>
> No, we did not implement the privacy auditing frameworks to empirically estimate the privacy loss of the algorithms. Not implementing privacy auditing is standard for existing papers (e.g., in Bu et al., Li et al., De et al., Bao et al., Yu et al., Tramet & Boneh), and we follow the same practice. Since our algorithm is aimed at improving the estimation quality of the privatized gradient, rather than improving the implementation of the privacy-preserving mechanism, we believe that privacy auditing is irrelevant to our algorithm. Additionally, as shown in the privacy auditing paper (e.g., Steinke et al. 2023), these auditing approaches tend to underestimate the privacy budgets (see Fig. 6 in Steinke et al.). Therefore, we do not conduct privacy auditing.
>
> However, we want to clarify that we adapt the public code bases of the previous SOTA results with minimum modification by adding the DiSK optimizer wrapper. Our code does not modify any part of the privacy accountant, gradient sampling, clipping, or other privatizing procedures of these code bases. Therefore, the privacy guarantees of our implementation and existing results should be exactly the same.

---

> ### Author Response · Authors · 2024-11-22
> **Response to Reviewer PnV2 Part II**
>
> > * The algorithms is 2 changes away from vanilla DP-SGD. The paper has hyperparmaeter tuning studies that are named ablation studies, but it does not have true ablation studies. There are 4 possible comparisons that can shed light on what is responsible for most of the apparent performance improvements. (1) vanilla dp-SGD (k=1, gamma=0); (2) weighted average before clipping, but no momentum on the gradient after clipping; (3) only momentum in the gradient and no pre-clipping averaging (gamma=0), (4) DiSK.
> >     * Corollary 1 implies that gamma may be the more important parameter.
>
> We agree that it is inaccurate to call it an "ablation study." The result in Fig. 8 demonstrates the trend and importance of $\kappa$ and $\gamma$, which "interpolates" between vanilla DPSGD and DiSK to some extent.
>
> For the "true" ablation study, we refer the reviewer to Fig. 6. In Fig. 6, we compared (1) vanilla DPSGD ($\kappa=1, \gamma=0$), (3) DPSGD with a low-pass filter (LP) that corresponds to only momentum in the gradient ($\kappa<1, \gamma=0$) and (4) DiSK (KF) ($\kappa<1, \gamma\neq 0$). We observe that applying the momentum improves the performance by a small amount, whereas DiSK has a much larger improvement. This indicates that the "weighted average before clipping" may be more important (i.e., $\gamma \neq 0$ is more important). We have revised the manuscript to clarify this point.
>
> As for case (2) $\kappa=1, \gamma \neq 0$, the algorithm reduces to vanilla DPSGD, and our theoretical algorithm design does not support only using weighted average before clipping. However, we agree that case (2) could be an interesting algorithm design.
>
> > * The paper does not explain how hyperparameters were selected. Were they chosen based on validation data (good) or testing data (thus favoring any approach that has more hyperparameters).
>
> We refer the reviewer to Appendix C.2, where we provide the hyper-parameter choices. The "optimal" hyper-parameter is chosen based on the test set (for CIFAR and MNIST, which do not have a validation set) and the validation set (for other datasets).
>
> > * I tried to match up some of the SOTA numbers in the appendix as a spot check.
> >     * Specifically, I looked at De et al. because they have a few prior SOTA numbers in the same place. I couldn't match their Imagenet numbers (32.4% top-1 accuracy to the 36.5 written in Appendix C5).
> >     * I also noticed that some of the high CIFAR-10 training from scratch numbers in De et al. are completely missing from the Appendix.
> >     * It is not clear to me whether DiSK is appropriately compared to the prior SOTA. Do you match deltas? Do you match neighbor definitions and batch subsampling? Do you match privacy accountants?
> >     * Delta is a big difference maker, so you should put its numerical value(s) in every table.
> >     * Table 6 does not match imagenet for table 2.
>
> Thank you for your detailed review. Let us provide more detail regarding  your concerns on the implementation and comparison.
>
> * **ImageNet-1k number:** As we explained in Appendix C.3, Lines 1277-1446, and in Fig. 7, the authors of De et al. have improved their [codes](https://github.com/google-deepmind/jax_privacy) (with various engineering add-ons compared with Fig. 2(c\)). Therefore, to have a fair comparison, we re-ran this experiment to get their improved number. We confirmed our results with one of the author's of the paper for a fair comparison. In the re-run, we achieve 40%, while their new result is 36.35% val. accuracy as reported in Fig. 7. We believe that this comparison is more accurate and fair than directly comparing with their old numbers (since we built our algorithm on top of their public codebase, which has already been improved from the publication date of their paper).
>
> * **Missing CIFAR-10 results in Tab. 6:** In Tab. 6, we only provide the results where DiSK improves SOTA (see Line 1416). Since our results in CIFAR-10 do not improve SOTA, we do not list them in the table. We have changed the caption of Tab. 6 to clarify this point further.
> * **For fair comparison:** the definition of the neighboring dataset is the same; we use the RDP privacy accountant; we provide the correct privacy guarantee; the $\delta$ used in the paper is smaller than SOTA; We don't see a reason why sampling and accountant need to be exactly the same. As long as the privacy guarantee is correct, we think the comparison is fair. In certain experiment settings, we re-ran the experiments (e.g. Imagenet-1k). We provide the link to our code in the original paper, line 1209, for examination.

---

> > ### Comment · Reviewer_PnV2 · 2024-11-22
> > **For fair comparison**
> >
> > In response to "We don't see a reason why sampling and accountant need to be exactly the same"
> >
> > Accountants should be matched to make sure the improvement is due to the algorithm rather than a more accurate accountant. Matching sampling is more useful for ablation study purposes. In any case, I appreciate the clarification that DiSK was implemented in a variety of frameworks and did not affect the accounting and noise injecting code. That is a useful detail to emphasize in a paper.

---

> > > ### Author Response · Authors · 2024-11-26
> > > **Thank you for raising your score**
> > >
> > > Thank you for reviewing our rebuttal and revising your score. We sincerely appreciate your thoughtful feedback, which has enhanced the quality and clarity of our paper.

---

> ### Author Response · Authors · 2024-11-22
> **Response to Reviewer PnV2 Part III**
>
> * **Listing $\delta$:** As stated in Sec. 5.1 in Hyper-parameter choice, we fix $\delta = 1/N^{1.1}$. Let us list the corresponding $\delta$'s used:
>
>   | Dataset  | Ours $\delta$ | SOTA $\delta$ |
>   | -------- | -------- | -------- |
>   | MNIST    | $5.5\times 10^{-6}$ | $10^{-5}$ |
>   | CIFAR-10/100 | $6.8 \times 10^{-6}$ | $10^{-5}$ |
>   | Imagenet-1k | $1.9\times 10^{-7}$ | $8\times 10^{-7}$   |
>   | MNLI | $6.3\times 10^{-7}$ | $1.1\times 10^{-6}$ |
>   | QNLI | $4.8\times 10^{-7}$ | $9\times 10^{-7}$ |
>   | SST-2| $4.9\times 10^{-6}$ | $7.4\times 10^{-6}$ |
>   | QQP  | $7.6\times 10^{-7}$ | $1.4\times 10^{-6}$ |
>   | E2E  | $1.2\times 10^{-5}$ | $1.2\times 10^{-5}$ |
>   | DART | $6.1\times 10^{-6}$ | $6.1\times 10^{-6}$ |
>
>   We have added it to our revised manuscript.
>
> ### Other questions/comments
> > * What is that wacky green line in Figure 1?
>
> As  mentioned in the legend, it demonstrates the amount of improvement of DiSK compared with the base optimizer. We clarified this point further in the caption in the revised manuscript.
>
> > * The paper uses C for too many different purposes. In some cases it is a clipping threshold, other times a constant in theoretical bounds, and other times a constant on properties of the data. This can get confusing even with subscripts.
>
> This is a great point. We will change the constants to A's.
>
> > * Line 121: where is the delta < 0.05 restriction coming from? If I am not mistaken, Definition 2 is all Balle & Wang and does not use the results from Dwork and Roth.
>
> Thanks for pointing out this issue. The limitation of $\delta$ should be $\delta \in [0,1]$ as in Balle & Wang. We refer to  Dwork and Roth because they proposed the Gaussian mechanism. Notice that we did not use this result in any of our analyses.
> > * Line 086-089: such comparisons are incomplete without specifying privacy parameters and adding citations.
>
> Great point! Initially, we wanted to avoid making the statements too long. But we agree with you that our space saving plan raised issues. We will modify the statement by adding citations and privacy budgets.

---

### Official Review · Reviewer_j3TA · 2024-11-01

**Soundness:** 3
**Presentation:** 2
**Contribution:** 2
**Rating:** 6
**Confidence:** 2

**Summary:**

This paper proposes using a Kalman filter to enhance the privacy-accuracy trade-off of DP optimizers. While traditional Kalman filtering introduces significant computational and memory demands, the authors provide a simplified approach that reduces these costs while retaining its advantages. Theoretically, they show that their method achieves better iteration complexity than DP-SGD by a constant factor. Experimentally, their method outperforms DP-SGD on various datasets.

**Strengths:**

- This paper effectively applies Kalman filtering from control theory to improve DP optimization algorithms, presenting an interesting connection between DP optimization and control theory.
- The algorithm improves upon the traditional Kalman filter and requires only a constant factor more space and computation than DP-SGD.
- The method is versatile and can be integrated with various DP optimization techniques, not just DP-SGD.

**Weaknesses:**

- The privacy analysis of algorithm 3 is not clear to me. Is it possible to express $\sigma_{DP}$ explicitly in terms of other variables?

- While the conditions in Theorem 2 and Figure 8 offer some guidance on hyperparameter selection, the interactions between different hyperparameters ($\gamma, \epsilon$ and $\kappa$) remain complex and unclear, making practical hyperparameter tuning challenging.

**Questions:**

See weakness.

---

> ### Author Response · Authors · 2024-11-22
> **Response to Reviewer j3TA**
>
> Thank you for recognizing our contribution and providing invaluable feedback to us. Below is our response to your questions:
>
> **Summary:**
> > ... Experimentally, their method outperforms DP-SGD on various datasets.
>
> We are glad that you found the numerical experiments strong. We would like to highlight that our comparisons extend beyond just DP-SGD. They include a range of algorithms, such as DP-SGD, DP-Adam, DP-AdamW, DP-LoRA, and the DP optimization approach proposed by De et al., incorporating various enhancements. We are excited that, in all cases, DiSK consistently outperforms various baseline DP optimizers.
>
> **Weaknesses:**
> > * The privacy analysis of algorithm 3 is not clear to me. Is it possible to express $\sigma_{DP}$ explicitly in terms of other variables?
>
> Thank you for raising this important question. The privacy analysis for DiSK is similar to that of DPSGD. An explicit (yet not tight) expression would be from Zhao et al. 2019: *for $\epsilon>0, \delta\leq 0.05$, $\sigma_{DP}=\Delta_\mathcal{A}(\frac{\sqrt{2\ln(1/2\delta)}}{\epsilon}+\frac{1}{\sqrt{2\epsilon}})$.* However, this is a loose bound and the result in Thm. 1 is a tight result.
>
> Let us further clarify why DiSK shares the same privacy analysis as DPSGD: Let us define a modified loss function:
> $$\tilde{f}(\mathbf{x};\gamma,\kappa,\mathbf{d},\xi) = \frac{1-\kappa}{\kappa\gamma}f(\mathbf{x}+\gamma\mathbf{d}\;\xi)+ (1-\frac{1-\kappa}{\kappa\gamma})f(\mathbf{x};\xi),$$
>
> Then, the privatized gradient query of DiSK (Alg. 3, Line 5) becomes
> $$\mathbf{g}_t = \frac{1}{B}\sum\_{\xi\in\mathcal{B}_t}\mathrm{clip}(\nabla\tilde{f}(\mathbf{x}_t; \gamma, \kappa, \mathbf{d},\xi), C)+\mathbf{w}_t,$$
> which applies the same sub-sampled Gaussian mechanism used in DPSGD (Alg. 1).
>
>
> > * While the conditions in Theorem 2 and Figure 8 offer some guidance on hyperparameter selection, the interactions between different hyperparameters ($\gamma, \epsilon$, and $\kappa$) remain complex and unclear, making practical hyperparameter tuning challenging.
>
> DiSK introduces only two extra hyper-parameters ($\gamma$, $\kappa$). Notice that $\epsilon$ is the privacy parameter and is not a tunable hyper-parameter.
>
> Tuning $\gamma, \kappa$ is not challenging. The results in Fig. 8(a) demonstrate a clear trend in the combinations of $\kappa, \gamma$, and (b) demonstrate its consistency for different $\epsilon$ values. We tune $\kappa,\gamma$ in different tasks and have the same observation. **We use $\kappa=0.7,\gamma=0.5$ in all experiments,** and, to our knowledge, this set of hyper-parameters provides universal good performance on all tasks.

---

> ### Comment · Reviewer_j3TA · 2024-11-22
>
> Thank you for addressing my questions. I have raised my score accordingly. However, I would greatly appreciate it if the authors could clarify the privacy accounting and improve the overall readability of the draft.

---

> > ### Author Response · Authors · 2024-11-26
> > **Additional clarification on the privacy accountant**
> >
> > We appreciate your valuable and prompt feedback. We would like to address your additional comments as follows:
> >
> > > I would greatly appreciate it if the authors could clarify the privacy accounting and improve the overall readability of the draft.
> >
> > We use the Renyi differential privacy (RDP) accountant in the Opacus and FastDP packages to numerically calculate the required injected DP noise $\sigma_{DP}$ to the gradient for fixed $(\epsilon,\delta)$-DP budgets. This accountant is the same as the ones used in the SOTA results, including De et al., Li et al., and Bu et al.. The detailed derivation of the RDP accountant can be found in [Wang et al.](https://proceedings.mlr.press/v89/wang19b/wang19b.pdf).
> >
> > We have added the clarifications in Appendix C.1 in the revised manuscript.

---

### Official Review · Reviewer_Nhmw · 2024-11-03

**Soundness:** 3
**Presentation:** 2
**Contribution:** 3
**Rating:** 6
**Confidence:** 3

**Summary:**

The paper proposes to use the Kalman filter for denoising gradient computation in differentially private optimization. Specifically, the Kalman filter views the gradient vector as the state and derives the system update equation based on the Taylor expansion of the gradient. The observation is then written as Gaussian perturbation of the gradient. However, naive incorporation of the Kalman filter involves computing the Hessian matrix, which is computationally infeasible for large models under DP constraints. To address the computational challenge, the authors proposed to use a simplified Kalman filter with noisy input, multiplicative observation noise, and diagonal covariance matrix. Theoretical analysis shows that the proposed algorithm satisfies $O\left(\sqrt{d/T}\right)$ convergence rate for smooth loss function with bounded per-sample gradient and bounded per-sample gradient variance, which has a constant factor improvement over DP-SGD under certain assumptions. Experiments validate the improvement of the proposed algorithm compared to DP-SGD and DP-Adam over a variety of DP vision and language tasks.

**Strengths:**

- The paper proposed an interesting way of viewing gradient update in differentially private training as a linear dynamics system. The proposed algorithm provides a novel and computationally efficient way of (approximately) incorporating the Kalman filter into DP training for denoising gradient.
- Extensive numerical experiments that validate the effectiveness of the proposed method.

**Weaknesses:**

- The reason for the gain of the proposed denoising algorithm is not perfectly clear from the presentation -- could the authors elaborate more on (1) why does the Kalman filter (as described in Algorithm 2) effectively denoise the gradient observation? (E.g., what is the minimized objective under denoising), (2) what are the assumptions for the simplifications made in Algorithm 3 and why are the assumptions meaningful?
- The authors mention that the algorithm performs better than all SOTA methods, including the DP-FTRL algorithm. However, DP-FTRL is also effectively a gradient denoising method that is proved optimal in terms of l2 error for releasing linear queries (gradient estimation in the context). Consequently, it is not very clear why the proposed Kalman filter method would be beneficial compared to another denoising method, and how much the improvement is.

**Questions:**

1. In Algorithm 2, are $\mathbb{E}[C]$ and $H_t$ data-dependent? If so, does Algorithm 2 satisfy differential privacy?

2. In Figure 2 (b), both the proposed algorithm KL-DPAdam and the baseline DP-Adam seem to achieve lower test accuracy than the SOTA results in De et al. 2022 Figure 1 (a). Does this contradict the claim that the proposed algorithm enables improvement compared to SOTA?

---

> ### Author Response · Authors · 2024-11-22
> **Response to Reviewer Nhmw Part I**
>
> We are glad that you found our approach novel and our experimental results strong. Below, we respond to your concerns and questions:
>
> **Weaknesses:**
> > * The reason for the gain of the proposed denoising algorithm is not perfectly clear from the presentation -- could the authors elaborate more on
> >   1. why does the Kalman filter (as described in Algorithm 2) effectively denoise the gradient observation? (E.g., what is the minimized objective under denoising),
> >   2. what are the assumptions for the simplifications made in Algorithm 3 and why are the assumptions meaningful?
>
> 1. **The objective of the Kalman filter:** In general, the Kalman filter aims to minimize the *mean square error* of the estimate of gradient given its noisy update and observation. For linear systems (e.g., for quadratic problems with inactive clipping in our context), the Kalman filter is known to be the best linear unbiased estimator (BLUE). Indeed, this is an important point, and we have included this discussion in Appendix A due to space limitations. We have highlighted it in the revised manuscript.
> 2. **Assumptions for simplification:** We made the following assumptions for simplification:
>    1. Observation matrix $\mathbf{C}_t = \mathbf{I}_d$.
>    2. Covariance matrix $\Sigma_H = \sigma^2_H\mathbf{I}_d$.
>
>    We would like to clarify that the simplification is used to design an executable algorithm with reasonable memory and computational cost. The assumptions made during the simplification in Sec. 3.2 are not theoretical justified, and the simplification of the Kalman filter deviates from minimizing MSE. However, the simplified algorithm we derived still have a number of desirable properties, for example, it still possesses strong theoretical properties as outlined in Sec. 4, and these properties **do not** rely on the assumptions made during the simplification.
>
> > * The authors mention that the algorithm performs better than all SOTA methods, including the DP-FTRL algorithm. However, DP-FTRL is also effectively a gradient denoising method that is proved optimal in terms of l2 error for releasing linear queries (gradient estimation in the context). Consequently, it is not very clear why the proposed Kalman filter method would be beneficial compared to another denoising method, and how much the improvement is.
>
> **Comparison with DP-FTRL:**
> DP-FTRL aims to minimize the DP noise injected to the sum of the privatized gradient sequence, and their claim is *the best known population risk guarantee for a single pass algorithm for convex problem*. In this framework, the gradient sequence is treated as an *arbitrary sequence of vectors*, and the algorithm aims to reduce the noise needed for privatization of the summation in an online fashion. However, in practice, gradient sequences are not arbitrary; they follow specific dynamics dictated by the optimization problem.
>
> In contrast, DiSK leverages the gradient dynamics over iterations to reduce the MSE of gradient estimation. The proposed algorithm can be applied to both convex and non-convex problems and a wide range of DP optimizers.
>
> Because of the differences in problem setting between DP-FTRL and DiSK, a direct comparison of the theoretical results is challenging. However, DiSK offers several advantages over DP-FTRL:
> 1. DiSK has less memory and computation cost. The memory and computation of DP-FTRL scales with at least $O(\ln(T)d)$ to store and compute correlated noise in a binary tree; band matrix factorization requires $O(n_bd)$ memory to store $n_b\sim400$ bands; and recent BLT approach requires $O(n_md)$ memory for $n_m\sim5$ buffers. DiSK only requires $2d$ extra memory.
> 2. DiSK leverages the gradient dynamics to improve DP optimization while DP-FTRL can be applied to any DP prefix sum releasing problem. It is not specially designed for DP optimization and does not leverage the structure in the gradient sequence.
> 3. DP prefix sum releasing problem, which is the core of DP-FTRL, is relevant in DP-SGD (where the final iterates is the summation of previous gradients). However, many optimizers (such as ADAM) do not follow this structure. In contrast, DiSK's approach is independent of the underlying optimization method, making it broadly applicable.

---

> ### Author Response · Authors · 2024-11-22
> **Response to Reviewer Nhmw Part II**
>
> **Questions:**
> > 1. In Algorithm 2, are $E[C]$ and $H_t$ data-dependent? If so, does Algorithm 2 satisfy differential privacy?
>
> Yes, they are data-dependent and Alg. 2 may not satisfy DP (depending on how we estimate $H_t$ and $E[C]$). That is also a reason why we simplify it to DiSK to ensure computation&memory efficiency and privacy. We clarified this point in our revised manuscript.
>
> > 2. In Figure 2 (b), both the proposed algorithm KL-DPAdam and the baseline DP-Adam seem to achieve lower test accuracy than the SOTA results in De et al. 2022 Figure 1 (a). Does this contradict the claim that the proposed algorithm enables improvement compared to SOTA?
>
> We want to clarify that the results in Fig. 1(a) in De et al. 2022 are the results for CIFAR-10, while Fig. 2(b) in this paper is for CIFAR-100. We provide a summary of the settings we improve on SOTA in Appendix C, Tab. 6, where we do not claim improving the SOTA on CIFAR-10 (for $\epsilon=1,4,8$).

---

> ### Comment · Reviewer_Nhmw · 2024-11-26
>
> Thanks for the clarifications, which address most of my concerns.
>
> > We want to clarify that the results in Fig. 1(a) in De et al. 2022 are the results for CIFAR-10, while Fig. 2(b) in this paper is for CIFAR-100. We provide a summary of the settings we improve on SOTA in Appendix C, Tab. 6, where we do not claim improving the SOTA on CIFAR-10 (for $\varepsilon=1, 4, 8$).
>
> Could the authors comment on why there is improvement for CIFAR-100 but not CIFAR-10? Is it a general trend that DISK is better suited for problems with a higher number of classes?

---

> ### Author Response · Authors · 2024-11-26
> **Response to the additional comment from Reviewer Nhmw**
>
> > Could the authors comment on why there is improvement for CIFAR-100 but not CIFAR-10? Is it a general trend that DISK is better suited for problems with a higher number of classes?
>
> In short, **the training approach in SOTA is tailored for CIFAR-10, while our training is not. DiSK improves most base algorithms.** Even with non-tailored training, DiSK improves SOTA results.
>
> Let us explain in details:
> 1. **CIFAR-10's SOTA results use optimized training procedure.** The SOTA results of CIFAR-10 for $\delta=10^{-5}$ and $\epsilon=1,2$ are $60.3\\%, 67.2\\%$ (Tramèr and Boneh, 2021) and use *handcrafted features*, for $\epsilon=4,8$ are $73.5\\%, 81.4\\%$ (De et al, 2022) and use *data augmentation, weight standarization and exponential moving averaging*. In comparison, we only combine DiSK with DP-Adam without using any advanced tricks, and we use smaller $\delta = 6.8\times10^{-6}$ for a tighter privacy guarantee. Therefore, we are not improving SOTA of CIFAR-10 for $\epsilon=1,4,8$. However, we still improve CIFAR-10's SOTA for $\epsilon=2$ to $68.8\\%$.
>
> 2. **Improving CIFAR-100 SOTA.** The tricks used in CIFAR-10 do not provide good result on CIFAR-100, see Bao et al. 2024. In contrast, DiSK still has a large improvement and outperforms SOTA with only running on top of DPAdam as the base algorithm.
>
> 3. **Improving base algorithms.** In training on the ImageNet-1k dataset in Appendix C3, Fig. 7, we apply exactly the same training procedure as in SOTA of CIFAR-10 and ImageNet-1k (i.e., De et al.) and demonstrate performance improvement. In the NLP results, we also apply DiSK to the same training procedure used in SOTA, and DiSK improves their performance. In these experiments, DiSK uses the **same set** of hyper-parameters as the base algorithms without further parameter tuning.
>
> In summary, we believe that DiSK is suitable for all problems (at least the settings we tried in our paper) and improves the performance of a wide range of base DP optimizers (and the ones used in SOTA) without re-tuning the hyper-parameters.
>
> We hope the above response can address your concern. Please let us know if you have any other questions.

---

### Official Review · Reviewer_pdcW · 2024-11-06

**Soundness:** 3
**Presentation:** 3
**Contribution:** 4
**Rating:** 8
**Confidence:** 3

**Summary:**

This paper introduces DiSK, a novel framework for enhancing differentially private (DP) optimizers by applying Kalman filtering to denoise privatized gradients, producing refined gradient estimates with reduced noise interference. By simplifying Kalman filtering for large-scale tasks, DiSK maintains efficiency and improves privacy-utility trade-offs, achieving significant performance gains over standard DP optimizers like DPSGD. Experimental results on vision and language tasks demonstrate DiSK's superiority under equivalent privacy constraints, achieving state-of-the-art performance across multiple benchmarks.

**Strengths:**

Empirical results are very strong and very significant. Computational costs (e.g. time/memory/parallelizability) of the method are quite good too. You mention this in theory, it may be good to make even a small empirical statement about this as well.

**Weaknesses:**

Why don’t people use Kalman filters for non-DP optimization? I don’t see why the same tricks you employed to not incur d^6 computational costs are not applicable to other non-private settings.
This feels like an important consideration to understand for contextualizing the empirical claims made in this work

**Questions:**

I’m very impressed with these results and somewhat surprised about why they work so much better than existing methods. In developing new methods, or building on top of this, what parts of this method can be extended - what are they key things that differentiate this from existing methods that don’t work as well. This may seem a bit naive, but my question is effectively - based on how big of an improvement this algorithm produces, what should next steps focus on?

Comment: I did not go through the utility proofs in great detail and could have missed something there.

---

> ### Author Response · Authors · 2024-11-22
> **Response to Reviewer pdcW**
>
> Thank you for recognizing the contributions of our work. We address your concerns and questions below:
>
> **Weaknesses:**
> > Why don't people use Kalman filters for non-DP optimization? I don't see why the same tricks you employed to not incur d^6 computational costs are not applicable to other non-private settings. This feels like an important consideration to understand for contextualizing the empirical claims made in this work
>
> This is an excellent question. General Kalman filtering has indeed been applied in non-DP optimization, as we noted in the paper. The simplified version derived in our work also connects to NAG and STORM algorithms, which we discuss in detail. Specifically, NAG and STORM can be interpreted as special cases of DiSK for particular choices of hyper-parameters.
>
> We did several limited experiments with DiSK in a few large-scale non-DP applications and observed minor benefits, though not as pronounced as in the DP setting. One possible explanation is that the Kalman filter performs best when the noise closely follows a Gaussian distribution—a condition that aligns well with DP optimization. Additionally, for the filter to function optimally, the system update noise should be smaller than the observation noise (ensuring $\kappa \neq 1$). In non-DP training, however, it's less clear whether the noise in SGD conforms to a Gaussian distribution or whether the system update noise (from Hessian estimation) is consistently smaller than the stochastic gradient noise.
>
> **Questions:**
> > I'm very impressed with these results and somewhat surprised about why they work so much better than existing methods. In developing new methods, or building on top of this, what parts of this method can be extended - what are the key things that differentiate this from existing methods that don't work as well. This may seem a bit naive, but my question is effectively - based on how big of an improvement this algorithm produces, what should next steps focus on?
>
> We are also very excited about our results, and we are glad that you found our work intriguing. Two key features of our framework that set it apart from some other recent developments, such as DP-FTRL, are as follows: 1) Our framework operates by postprocessing gradients, allowing it to be seamlessly integrated with any optimizer. As long as the optimizer accepts gradients as input, DiSK can be employed to enhance its performance. 2) Unlike approaches that treat gradients as non-related entities (e.g., DP-FTRL), our framework leverages the underlying structure of the optimization problem. Specifically, it utilizes the relationship between (the consecutive) true gradients to denoise them more effectively.
>
> While we are excited about our current results, there is a lot to do as follow-up works. For example, we do not yet account for the non-linearity introduced by the multiplicative noise of clipping and subsampling in DiSK's updates. Additionally, DiSK requires more memory compared to non-denoised algorithms, which can pose challenges when training or fine-tuning very large models or in memory-constrained environments. Lastly, as you pointed out, applying, studying, and modifying DiSK for non-DP training requires further investigation.

---

> > ### Comment · Reviewer_pdcW · 2024-11-25
> >
> > Great, thank you. As my score was originally high, I leave it unchanged.

---

### Meta-Review · Area_Chair_4peK · 2024-12-21

**Metareview:**

This paper contributes to a growing body of work that aims to improve the performance (utility) of differentially private stochastic gradient descent (DP-SGD) by post-processing gradients. The main contribution is DiSK: a simplified Kalman Filter to denoise privatized gradients. The diSK is theoretically grounded and demonstrates favorable performance relative to competing state-of-the-art methods in various benchmarks.

The reviewers noted several positive points in the paper, including "strong and significant empirical results" and "Computational costs (e.g. time/memory/parallelizability) of the method are quite good too" (pdcW), the method's novel and computationally efficient way of (approximately) incorporating the Kalman filter into DP training (Nhmw), and the algorithm being practically feasible and theoretically grounded (PnV2).

The reviewers also noted several areas of improvement, including ambiguity in the experimental setups and comparisons to prior SOTA and discrepancies in experimental results (PnV2), clarity and presentation of the DiSK algorithm (Nhmw). They noted that the privacy analysis could be streamlined (j3TA).

Overall, this is a nice contribution of practical relevance that pushes the utility of DP-SGD forward -- a clear acceptance. I encourage the authors to add the promised changes in the final version of the paper.

**Additional Comments On Reviewer Discussion:**

The initial reviews posed several clarifying questions to the authors, which were appropriately addressed in the rebuttal. The reviewers unanimously agreed on the paper's acceptance.

---

### Decision · Program_Chairs · 2025-01-22

Accept (Poster)